# Molecular pathways of senescence regulate placental structure and function

Hilah Gal[1], Marina Lysenko[2], Sima Stroganov[1], Ezra Vadai[1], Sameh A Youssef[3,4],
Keren Tzadikevitch-Geffen[5], Ron Rotkopf[6], Tal Biron-Shental[5], Alain de Bruin[3,4] (iD), Michal Neeman[2] &
Valery Krizhanovsky[1,*] (iD)

## Abstract

The placenta is an autonomous organ that maintains fetal growth and development. Its multinucleated syncytiotrophoblast layer, providing fetal nourishment during gestation, exhibits characteristics of cellular senescence. We show that in human placentas from pregnancies with intrauterine growth restriction, these characteristics are decreased. To elucidate the functions of pathways regulating senescence in syncytiotrophoblast, we used dynamic contrast-enhanced MRI in mice with attenuated senescence programs. This approach revealed an altered dynamics in placentas of $p53^{-/-}$, $Cdkn2a^{-/-}$, and $Cdkn2a^{-/-};p53^{-/-}$ mice, accompanied by histopathological changes in placental labyrinths. Human primary syncytiotrophoblast upregulated senescence markers and molecular pathways associated with cell-cycle inhibition and senescence-associated secretory phenotype. The pathways and components of the secretory phenotype were compromised in mouse placentas with attenuated senescence and in human placentas from pregnancies with intrauterine growth restriction. We propose that molecular mediators of senescence regulate placental structure and function, through both cell-autonomous and non-autonomous mechanisms.

Keywords gelatinase; intrauterine growth restriction; placenta; senescence; syncytiotrophoblast

Subject Categories Cell Cycle; Development & Differentiation

The EMBO Journal (2019) 38: e100849

## Introduction

Cellular senescence is defined as a stable cell-cycle arrest that limits the proliferative potential of cells. A senescence program can be induced in response to various forms of cellular stress, including oncogene activation, telomere shortening, oxidative stress, and other types of stress leading to DNA damage (Campisi, 2011; Munoz-Espin & Serrano, 2014; Salama *et al*, 2014; Childs *et al*, 2015). Cell–cell fusion, mediated by the ERVWE1 fusogen, can also trigger a senescence response in several tissue culture systems and in the placenta (Chuprin *et al*, 2013). Short-term effects of the senescence program are needed for tumor suppression, restriction of tissue damage, and embryonic development (Serrano *et al*, 1997; Braig *et al*, 2005; Chen *et al*, 2005; Collado *et al*, 2005; Michaloglou *et al*, 2005; Krizhanovsky *et al*, 2008; Chuprin *et al*, 2013; Munoz-Espin *et al*, 2013; Storer *et al*, 2013). In contrast, the long-term accumulation of senescent cells in tissues has detrimental consequences, such as promotion of tumorigenesis and age-related pathologies (Campisi, 2011; Munoz-Espin & Serrano, 2014; Ovadya & Krizhanovsky, 2014; Childs *et al*, 2015; Baker *et al*, 2016).

The senescence program is coordinately regulated by the p53/p21 and p16/pRb tumor-suppressor networks that promote and sustain growth arrest in senescent cells (Campisi, 2011; Burton & Krizhanovsky, 2014; Munoz-Espin & Serrano, 2014). In addition to the activation of molecular pathways that reinforce cell-growth arrest, a hallmark of senescent cells is the secretion of pro-inflammatory cytokines, chemokines, growth factors, and proteases, also referred to as the senescence-associated secretory phenotype (SASP; Acosta *et al*, 2008; Coppe *et al*, 2008; Kuilman *et al*, 2008; Wajapeyee *et al*, 2008). Activation of the SASP can, in a cell non-autonomous manner, modulate the senescent cell microenvironment, promote the interaction with the immune system, and reinforce growth arrest of the adjacent senescent cells (Acosta *et al*, 2008, 2013; Krizhanovsky *et al*, 2008; Kuilman *et al*, 2008; Lujambio *et al*, 2013). Regulation of the SASP is complex, and the expression of its components is governed in a coordinated manner by the DNA damage response and the NF-κB and JAK-STAT signaling systems (Acosta *et al*, 2008; Kuilman *et al*, 2008; Rodier *et al*, 2009; Chien *et al*, 2011; Xu *et al*, 2015). These pathways may also contribute to the tumor-suppressive role of senescence.

1   Department of Molecular Cell Biology, The Weizmann Institute of Science, Rehovot, Israel
2   Department of Biological Regulation, The Weizmann Institute of Science, Rehovot, Israel
3   Department of Pathobiology, Faculty of Veterinary Medicine, Dutch Molecular Pathology Center, Utrecht University, Utrecht, The Netherlands
4   Division of Molecular Genetics, Department of Pediatrics, University Medical Center Groningen, University of Groningen, Groningen, The Netherlands
5   Department of Obstetrics and Gynecology, Meir Medical Center, Kfar Saba, Israel
6   Bioinformatics and Biological Computing Unit, Department of Biological Services, The Weizmann Institute of Science, Rehovot, Israel
    *Corresponding author. Tel: +972 8 934 6575; E-mail: valery.krizhanovsky@weizmann.ac.il

In addition to its protective function in suppressing tumors and limiting tissue damage, senescence plays a physiological role in embryogenesis by instructing tissue growth and patterning of organs (Munoz-Espin *et al*, 2013; Storer *et al*, 2013). Developmentally programmed senescent cells are present, at different stages of embryonic development, in several transitory embryonic organs including the mesonephros, the endolymphatic sac of the inner ear, the apical ectodermal ridge, the neural roof plate, and several other tissues (Munoz-Espin *et al*, 2013; Storer *et al*, 2013). Activation of the developmental pathways of senescence seems to differ from the activation of senescence pathways triggered by DNA damage. Although these cells are strictly dependent on expression of p21 (Cdkn1a) via the transforming growth factor (TGF)-β/ SMAD and PI3K/FOXO pathways, they do not display a persistent DNA damage response and can be induced independently of p53 and p16. Interestingly, developmental senescent cells also share expression signatures with oncogene-induced senescence, especially those related to SASP (Storer *et al*, 2013). Therefore, senescent cells in embryos use SASP components to regulate temporal and spatial patterning.

During development, senescence pathways are also induced in the placenta (Chuprin *et al*, 2013). Unlike the signature of programmed developmental senescence in the embryo, senescence in the placenta shares features of senescence induced by DNA damage and exhibits a coordinated activation of p53/p21 and p16/ pRb regulatory pathways (Chuprin *et al*, 2013). ERVWE1, a fusogen of viral origin, is endogenously expressed in the placenta and mediates cell-fusion-induced senescence of the syncytiotrophoblast, following cytotrophoblast differentiation. This multinucleated cell type serves as a maternal–fetal interface, supporting fetal growth and development (Rossant & Cross, 2001; Georgiades *et al*, 2002; Cox & Redman, 2017). It develops and functions in a coordinated manner, together with placental vascular structures. In mice, the function of the syncytiotrophoblast/vascular interface can be monitored non-invasively *in vivo* by dynamic contrast-enhanced magnetic resonance imaging (DCE-MRI) (Plaks *et al*, 2011a,b). This technique has been successfully implemented in studies of tetraploid placental complementation, fetal growth, implantation, and mouse models of pregnancy complications including intrauterine growth restriction (IUGR; Plaks *et al*, 2011a,b; Solomon *et al*, 2014; Avni *et al*, 2015). In humans, IUGR is among the most frequent pregnancy complications associated with abnormalities in placental growth, structure, and function, and is a major cause of fetal morbidity and mortality (Regnault *et al*, 2002; Burton *et al*, 2009; Cox & Redman, 2017). The molecular and cellular mechanisms that lead to this deleterious pregnancy complication have yet to be elucidated.

Our aim in this study was to understand the role of molecular pathways of senescence on placental structure and function. To this end, we examined the expression of central mediators of senescence in the human pathology of IUGR and found strong alterations in their expression. To investigate the impact of senescence mechanisms on placental function, we performed DCE-MRI-based analysis of macromolecular biotin-BSA-GdDTPA dynamics in mouse placentas in which molecular pathways of senescence had been inactivated. We observed that signal intensity (SI) dynamics in placentas were significantly altered in cases where both of the parallel senescence regulators, *p53* and *Cdkn2a* (*p16/p19*), were disabled.

Histological analyses of these placentas clearly indicated an altered placental structure. To understand the molecular mechanism employed by senescence regulators in the placenta, we performed genome-wide transcriptional analysis of human primary syncytiotrophoblast cultures exhibiting multiple features of cellular senescence. We characterized the central mediators and regulatory pathways of senescence that participate in placental regulation and examined the functional and structural consequences when they were eliminated from murine models. Strikingly, the same pathways were found to be attenuated in the human pathology of IUGR.

# Results

## Senescence pathways are downregulated in the human placenta during IUGR pathology

We aimed to understand the contribution of senescence-associated pathways to placental function. To investigate the possible implication of senescence in pregnancy complications, we focused on IUGR, a pregnancy complication characterized by restricted fetal growth. Senescence pathways and regulators, including the CDK inhibitors p16 and p21, are activated in the syncytiotrophoblast of third-trimester placentas (Fig 1A and B; Chuprin *et al*, 2013). We therefore proceeded to determine whether the senescence pathways are altered in syncytiotrophoblast cells of placentas from IUGR-complicated pregnancies. Post-partum, we examined sections of third-trimester uncomplicated (normal) and IUGR human placentas of the same gestational age (Appendix Table S1), using immunohistochemistry for markers of senescence. Interestingly, the percentages of syncytiotrophoblast nuclei positive for the CDK inhibitors p15, p16, and p21 were significantly reduced in the IUGR compared with the normal placentas ($P < 0.001$, $P < 0.01$, and $P < 0.05$, respectively, Fig 1C–E). Consistently with the reduction in expression of the above markers, nuclear staining for p53 was also strongly reduced in syncytiotrophoblast of IUGR placentas ($P < 0.01$, Fig 1F). When examined by immunoblot analysis, expression of the senescence markers p16, p21 p53, and DCR2 was also reduced in the IUGR placentas compared with the normal placentas of the same gestational age (Fig 1G). Furthermore, RT–PCR analysis confirmed significant reductions in p16, p21, and DCR2 expression in IUGR compared with normal placentas (Fig 1H). Altogether, these findings indicated that the central pathways of senescence, p16-pRb and p53-p21, are dysregulated in syncytiotrophoblast cells of IUGR placentas.

## Molecular mechanisms of senescence affect placental DCE-MRI dynamics

The multinucleated syncytiotrophoblast layer of the placenta exhibits various markers of cellular senescence (Chuprin *et al*, 2013) that are dysregulated in the placental pathology of IUGR. However, the functional significance of senescence in syncytiotrophoblast for the mammalian placenta is not understood. We therefore set out to determine the impact of the molecular mechanisms of senescence on placental structure and function *in vivo*. To evaluate placental function, we studied pregnant mice using *in utero* DCE-MRI, using a 9.4 T Bruker scanner. Macromolecular DCE-MRI is a novel,

effective, and non-invasive method for functional *in utero* phenotypic evaluation of maternal circulation in the mouse placenta and has been successfully employed in various applications such as the study of fetal implantation and placental manipulations (Plaks *et al*, 2006, 2011b; Avni *et al*, 2015).

We evaluated placentas on embryonic day E14.5 of gestation, since previous studies of placentas deficient in syncytiotrophoblast formation have reported differences in placental architecture and embryo survival outcome on day E14.5 and later (Dupressoir *et al*, 2009, 2011). To evaluate placental circulation, we injected pregnant

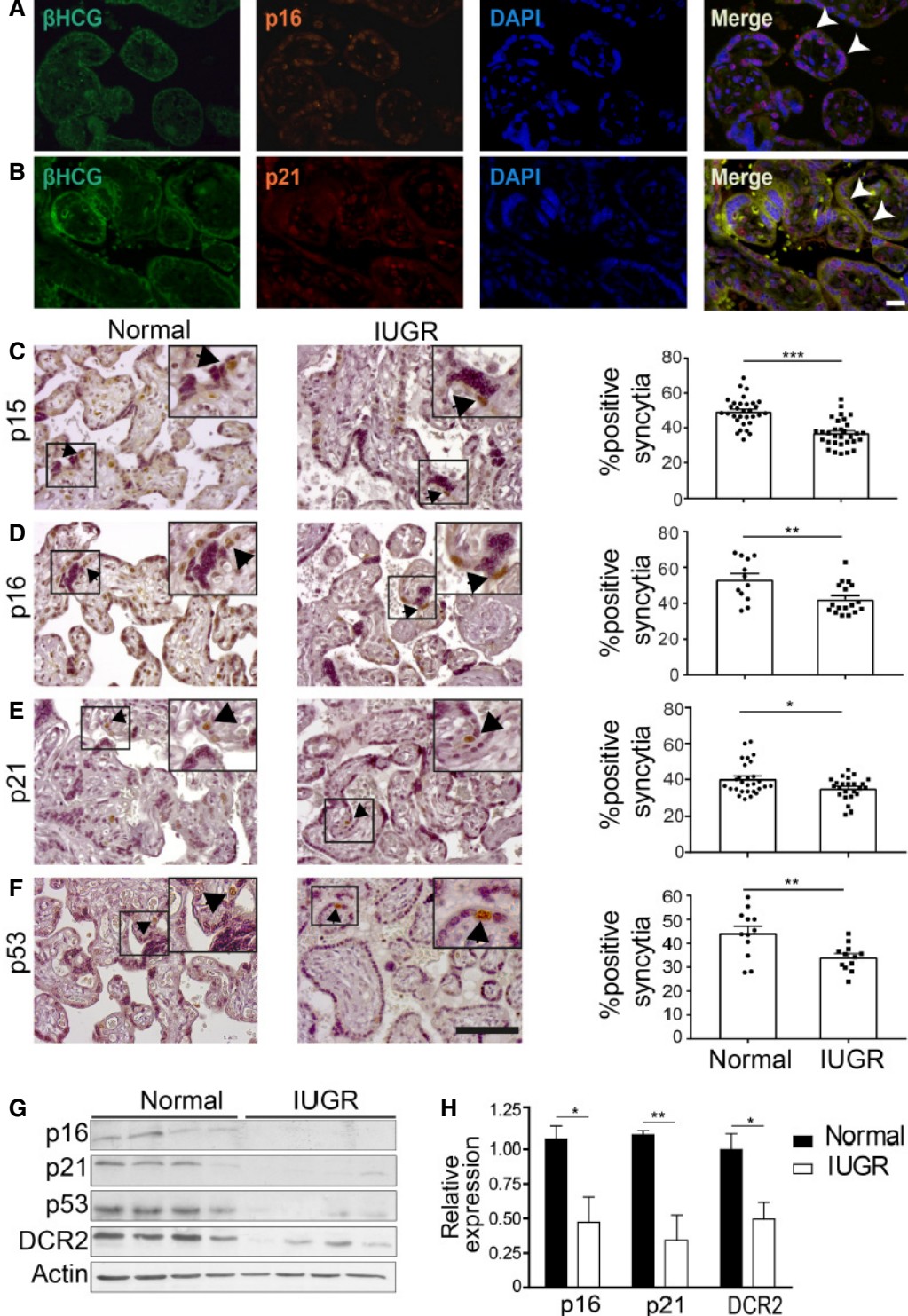

**Figure 1.**

**Figure 1. Senescence pathways are downregulated in human placentas complicated by IUGR.**

A, B    Human placental syncytiotrophoblast cells express markers of cellular senescence regulators, p16 and p21. Sections of normal human post-partum, third trimester, placenta were evaluated by immunofluorescence co-staining for senescence regulators (A) p16 and (B) p21. Green label: the syncytiotrophoblast marker βHCG, red label: p16 or p21, and blue label: DAPI nuclear stain. Arrowheads indicate positively stained syncytiotrophoblast cells βHCG⁺/p16⁺ (A) and βHCG⁺/p21⁺ (B). Scale bar, 50 μm.

C–F    Sections of human normal third-trimester placentas post-partum (n = 4 normal placentas) and IUGR-complicated placentas of the same gestational age (n = 4 IUGR placentas) were stained for senescence-related markers of p15 (C), p16 (D), p21 (E), and p53 (F) in the syncytiotrophoblast (n = 3 sections, derived from each normal and IUGR placenta). Percentages of positively stained syncytia were quantified (n = at least 12 fields of view from all sections). Scale bar, 100 μm. Arrowheads indicate syncytiotrophoblast cells.

G    Immunoblot analysis of the protein content of p16, p21, p53, and DCR in normal (n = 4 placentas) and in IUGR-complicated placentas (n = 4 placentas). Each lane represents one independent placenta.

H    Quantitative RT–PCR analysis of p16, p21, and DCR2 expression in IUGR-complicated and in normal placentas. Results were obtained from four human normal and four IUGR placentas.

Data information: Values are means + SEM of at least three experimental repeats. Statistical significance was determined by unpaired two-tailed Student's *t*-test
*P < 0.05; **P < 0.01; ***P < 0.001.

C57BL/6 wild-type (WT) mice intravenously with an albumin-labeled contrast agent (biotin-BSA-GdDTPA) and monitored them using *in utero* DCE-MRI on E14.5. DCE-MRI methodology successfully evaluated normal murine placentas at this developmental stage (Solomon *et al*, 2014). Owing to the high molecular weight of the contrast material, it does not cross the syncytiotrophoblast barrier into the fetus, thus allowing the maternal circulation to be analyzed within the placenta. This contrast agent was previously shown to localize in the labyrinth region of the placenta where the syncytiotrophoblast cells reside (Appendix Fig S1) and be taken up by trophoblast cells with resulting signal attenuation and contrast (Plaks *et al*, 2011a; Solomon *et al*, 2014). After injection of the contrast agent, we immediately followed up with DCE-MRI for 60 min (Fig 2A and C, Movie EV1). The region of interest of each placenta was marked (Fig 2A) and the mean SI in each placenta was calculated over time, as illustrated in a single placenta from a WT pregnant mouse (Fig 2B and C). The SI dynamics of the WT placenta showed enhancement of SI, followed by its reduction and a SI recovery stage (Fig 2B). This particular SI dynamics of initial enhancement is due to phagocytic internalization of the contrast agent by the labyrinth syncytiotrophoblasts and trophoblast giant cells (Plaks *et al*, 2011b). The "recovery stage" of SI enhancement is due to recycling of the biotinylated contrast agent back into the maternal circulation. Accordingly, we established a baseline SI dynamics in WT mice to serve as a basis for further comparisons.

To gain an understanding of the functional role of molecular mechanisms of senescence in the syncytiotrophoblast, we performed DCE-MRI on pregnant mice with attenuated senescence programs, using knockout mice for *Cdkn1a* (*p21*)*, p53, Cdkn2a*, and combined knockout of *Cdkn2a* and *p53*. We monitored the SI dynamics of placentas of *Cdkn1a*⁻/⁻ (Fig 2D)*, p53*⁻/⁻ (Fig 2E)*, Cdkn2a*⁻/⁻ (Fig 2F), and *Cdkn2a*⁻/⁻;*p53*⁻/⁻ embryos (Fig 2G) for 60–75 min, and compared them with placentas of WT or heterozygous (HET) embryos, derived from the same litters, on E14.5. To quantify the changes in SI dynamics, the SI parameters of initial enhancement and of recovery were defined as the ratios between the first SI maximum and the initial SI point, and between the second SI maximum and the SI minimum between the two maximum points, respectively (Fig 2B). We compared these parameters of SI dynamics on day E14.5 in WT, *Cdkn1a*⁻/⁻*, p53*⁻/⁻, *Cdkn2a*⁻/⁻, and *Cdkn2a*⁻/⁻;*p53*⁻/⁻ placentas in all combined experiments. SI dynamics of initial enhancement were altered in

placentas of *Cdkn2a*⁻/⁻;*p53*⁻/⁻ compared with WT mice (P < 0.01; Fig 2H). Similarly, compared to WT placentas, we observed a reduction in the recovery parameters in placentas of *p53*⁻/⁻ (P < 0.05) and *Cdkn2a*⁻/⁻ (P < 0.05), with the most significant reduction seen in *Cdkn2a*⁻/⁻;*p53*⁻/⁻ (P < 0.001; Fig 2I). We used WT placentas as the common denominator for this comparison because of the variability in SI among knockouts of different genotypes. Similar findings were obtained in the SI parameters for each genotype when compared to its WT or, in the case of *Cdkn2a*⁻/⁻;*p53*⁻/⁻, to HET (*Cdkn2a*⁻/⁻;*p53*⁺/⁻) littermates, owing that it is impossible to have WT littermates for this genotype (Appendix Fig S2). These data indicated that SI dynamics are significantly altered in placentas in which both senescence-regulating pathways, p53-p21 and p16-pRb, are disabled.

## Molecular mechanisms of senescence sustain placental morphology

Our present findings revealed that molecular mechanisms of senescence participate in maintaining normal placental function, as revealed by DCE-MRI dynamics. The observed changes in such dynamics might have resulted from structural changes in the placenta. We therefore proceeded to evaluate the impact of the senescence program on placental morphology, particularly in the labyrinth layer, which contains the syncytial cells. We examined the labyrinth region of WT, *Cdkn2a*⁻/⁻, *Cdkn2a*⁻/⁻;*p53*⁻/⁻, *p53*⁻/⁻, and *Cdkn1a*⁻/⁻ mouse placentas, on E14.5. Interestingly, in the labyrinth zones of *Cdkn2a*⁻/⁻ and *Cdkn2a*⁻/⁻;*p53*⁻/⁻ placentas we found structural abnormalities, including mild-to-moderate labyrinth trophoblast hyperplasia, collapsed vasculature, and nuclear enlargement of labyrinth trophoblasts (polyploidy), whereas no such abnormalities were detectable in WT placentas (Fig 3A–C). In addition to collapsed vasculature, we noted smaller, compressed, and unequal distribution of blood vessel lumina in the labyrinth of *Cdkn2a*⁻/⁻;*p53*⁻/⁻ compared with WT (Appendix Fig S3). Furthermore, *Cdkn2a*⁻/⁻;*p53*⁻/⁻ labyrinth exhibited more intense staining of the cytotrophoblast marker Epcam by immunofluorescence compared with WT, most likely illustrating the hyperplasia of the labyrinth trophoblasts (Appendix Fig S3). Interestingly, we also observed structural changes in the trophospongium zone of the *Cdkn2a*⁻/⁻;*p53*⁻/⁻ placentas which was significantly thicker and appeared more cellular compared with WT placenta (Appendix Fig

S4). In contrast to the phenotypic anomalies detected in the $Cdkn2a^{-/-}$ and the $Cdkn2a^{-/-};p53^{-/-}$ placentas, no significant changes were observed in the labyrinth regions of $p53^{-/-}$ (Fig 3D) or $Cdkn1a^{-/-}$ mice (Fig 3E), relative to WT placentas. In $Cdkn2a^{-/-}$ placentas, structural differences were readily noticeable and were

similar to those observed in $Cdkn2a^{-/-}:p53^{-/-}$ placentas. Thus, both $Cdkn2a^{-/-}$ and $Cdkn2a^{-/-}:p53^{-/-}$ placentas exhibit structural alterations.

Lack of proliferation and increased SA-β-gal activity are characteristic of the senescence phenotype. The syncytiotrophoblast in the

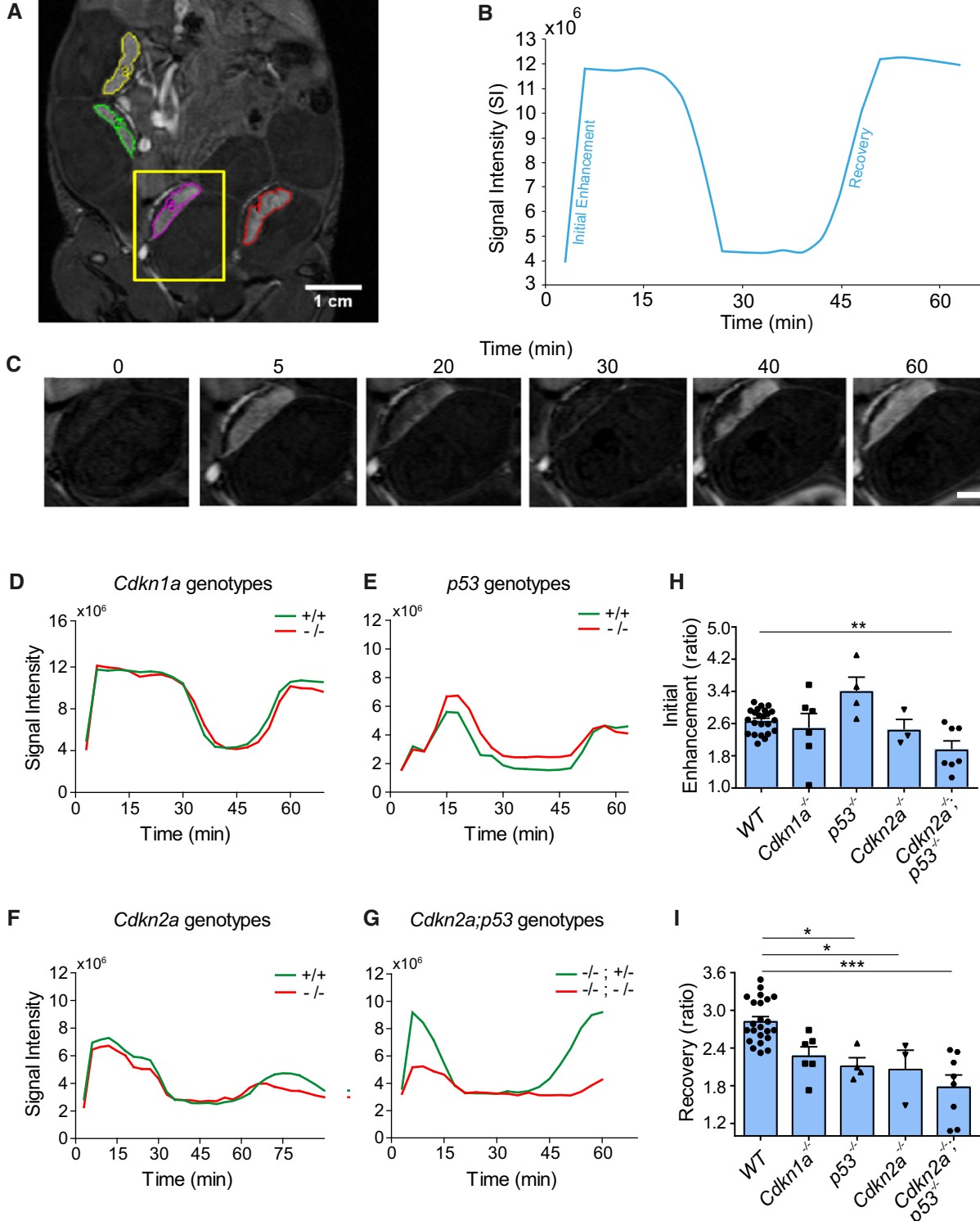

**Figure 2.**

◄

**Figure 2. Placentas with attenuated senescence programs exhibit altered signal intensity dynamics on *in utero* DCE-MRI.**

DCE-MRI was performed on pregnant mice of WT, *Cdkn1a*, *p53*, *Cdkn2a*, and *Cdkn2a;p53* genotypes on day E14.5. Mice were injected i.v. with an albumin-labeled contrast agent (biotin-BSA-GdDTPA; 10 mg/mouse), and placental enhancement was monitored at 9.4T MRI for 60–75 min.

A Representative T1-weighted image of a pregnant WT mouse and a selection of four different placentas (outlined by colored lines).

B Representative SI dynamics of a single WT placenta (marked in A), following biotin-BSA-GdDTPA administered for 60 min. SI parameters of initial enhancement and recovery are illustrated.

C Representative SI dynamics of a single WT placenta (marked in A) over time. Scale bar, 2.5 mm.

D–G Representative SI plots of *Cdkn1a*$^{-/-}$ (D), *p53*$^{-/-}$ (E), *Cdkn2a*$^{-/-}$ (F), and *Cdkn2a*$^{-/-}$;*p53*$^{-/-}$ (G) placentas, compared with their WT or HET littermates.

H, I Quantification of the SI initial enhancement (H) and SI recovery (I) in WT, *Cdkn1a*$^{-/-}$, *p53*$^{-/-}$, *Cdkn2a*$^{-/-}$, and *Cdkn2a*$^{-/-}$;*p53*$^{-/-}$ placentas. MRI experiments were repeated at least three times for each murine genotype ($n \geq 3$ placental SI measurements from each genotype). Values are means + SEM; statistical significance was determined by unpaired two-tailed Student's *t*-test *$P < 0.05$; **$P < 0.01$; ***$P < 0.001$.

labyrinth layer of the mouse placenta is formed by cell fusion (Rossant & Cross, 2001; Dupressoir *et al*, 2011), and expresses cellular senescence markers including SA-β-gal activity, p16, p53, and ARF, which are reduced in *Cdkn2a*$^{-/-}$:*p53*$^{-/-}$ placentas (Fig 3H, and Appendix Figs S5 and S6). To determine the levels of proliferation in the labyrinth zone of the placentas in which the syncytiotrophoblast cells reside, we performed immunohistochemical staining for Ki67 on placental sections of *WT*, *p53*$^{-/-}$, *Cdkn2a*$^{-/-}$, and *Cdkn2a*$^{-/-}$;*p53*$^{-/-}$ mice, on E14.5. The analysis of the Ki67 proliferation marker revealed its increase in more than two-fold in placentas of *Cdkn2a*$^{-/-}$ and *Cdkn2a*$^{-/-}$;*p53*$^{-/-}$ relative to WT placentas (Fig 3F and G). Overall, we found a strong association between a high proliferation level in the labyrinth zone of *Cdkn2a*$^{-/-}$ or *Cdkn2a*$^{-/-}$;*p53*$^{-/-}$ placentas and the increased hypercellularity observed in this region (Fig 3B and C). To determine whether the hyperproliferation originates from the labyrinth progenitor trophoblast cells, we performed immunofluorescence co-staining for Ki67 and the cytotrophoblast marker Epcam (Appendix Fig S7A). The image analysis revealed enhanced Ki67 expression in Epcam-positive cell population in the *Cdkn2a*$^{-/-}$;*p53*$^{-/-}$ placental labyrinth, compared with WT ($25.0\% \pm 1.22$ versus $17.9\% \pm 1.39$, for Cdkn2a;p53 knockout and wild-type, respectively; Appendix Fig S7B). This observation was further confirmed by immunoblot analysis of Epcam in *Cdkn2a*$^{-/-}$;*p53*$^{-/-}$ and WT placentas (Appendix Fig S7C). Moreover, the increased proliferation in the labyrinth zones of *Cdkn2a*$^{-/-}$ and *Cdkn2a*$^{-/-}$;*p53*$^{-/-}$ placentas was accompanied by a marked reduction in SA-β-gal activity (Fig 3H and I). Although there were no significant changes in SA-β-gal activity in the labyrinth zone of *Cdkn1a*$^{-/-}$ and *p53*$^{-/-}$ placentas relative to WT (Appendix Fig S8), significant reductions in SA-β-gal activity (of 2.22- and 3.32-fold, respectively) were observed in the labyrinth of *Cdkn2a*$^{-/-}$ and *Cdkn2a*$^{-/-}$;*p53*$^{-/-}$, compared with *WT* placentas ($P < 0.001$ for both genotypes; Fig 3I). Altogether, we found that lack of senescence mediators, in particular *Cdkn2a*, or *Cdkn2a* in combination with *p53*, significantly affects placental SI dynamics by MRI, placental morphology in the labyrinth zone, proliferation and SA-β-gal activity.

## Human syncytiotrophoblasts in culture activate senescence pathways

Molecular markers of senescence are expressed in human syncytiotrophoblast, but the molecular mechanisms that govern senescence in these cells are not yet understood. To unravel the molecular mechanisms that mediate the impact of senescence on the placenta, we studied human syncytiotrophoblast in culture. In the mammalian placenta, the villous cytotrophoblast cell population is continuously incorporated by syncytial fusion into the placental syncytiotrophoblast layer (Georgiades *et al*, 2002; Li & Schust, 2015). We isolated cytotrophoblast cells from human term placentas. In culture, cytotrophoblast cells spontaneously differentiate into syncytium, creating the large multinucleated syncytiotrophoblast structure that can be monitored and studied for its molecular and cellular characteristics (Kliman *et al*, 1986; Li & Schust, 2015). To study the nature of the syncytiotrophoblast, we generated cytotrophoblast cultures and assessed them for purity of the trophoblast population by immunofluorescence, using the trophoblast marker cytokeratin-7 and the ERVWE1 fusogen. The fused syncytiotrophoblast showed positive expression of cytokeratin-7 on day 3 after plating (Appendix Fig S9A), confirming the efficient isolation of trophoblast cells compared with negatively stained IMR90 fibroblasts. The ERVWE1 fusion protein, which is responsible for fusion of cytotrophoblast cells into syncytiotrophoblast (Cox & Redman, 2017), showed positive expression on day 3 (Appendix Fig S9B), consistent with the observed fusion of the majority of trophoblast cells at that time (Fig 4A and B). To determine the rate of syncytiotrophoblast formation by cell fusion, cells were stained for the F-actin cytoskeletal protein and monitored at 1, 3, and 5 days after plating (Fig 4A and B, and Appendix Fig S9C). We found that trophoblast cell fusion occurred as early as 1 day after seeding, at which time it exhibited 40.5% of fused cells, with a gradual increase to 60 and 73% fused cells on days 3 and 5, respectively (Fig 4B). While fusion was already occurring 1 day after seeding, most of the fused cells had only two or three nuclei at this time point compared with the multinuclear cells seen on days 3 and 5 (Fig 4A and B, and Appendix Fig S9C). Furthermore, secretion of the human chorionic gonadotropin-beta (βhCG) hormone, normally secreted exclusively by syncytiotrophoblast cells (Appendix Fig S10), increased with time, indicating syncytiotrophoblast differentiation (Fig 4C). Altogether, we found that human primary trophoblast cells fuse in culture to form the syncytiotrophoblast, which expresses its characteristic physiological and molecular markers.

To gain an insight into the molecular mechanisms activated during fusion of the placental syncytiotrophoblast, we performed mRNA profiling on primary trophoblasts obtained independently from two human term placentas, at 1.5, 2, 3 and 5 days after seeding. The expression matrix revealed similar gene-expression patterns between 1.5- and 2-day cultures and between 3- and 5-day cultures (Appendix Fig S11). We identified 204 modulated genes, selected on the basis of a 1.8-fold change between 3- to 5-day cultures and 1.5- to 2-day cultures (Appendix Fig S11, Appendix Tables S2 and S3). Owing to the similarity of the

expression profiles obtained, we focused our analysis on gene expression on day 5 compared with that on day 2. Notably, on day 5 cultures the majority of trophoblast cells had already fused into a fully formed multinuclear syncytiotrophoblast layer (Fig 4A and B, and Appendix Fig S9C), as opposed to early cultures, in which most cells remain unfused, and the phenotype of syncytiotrophoblast, as

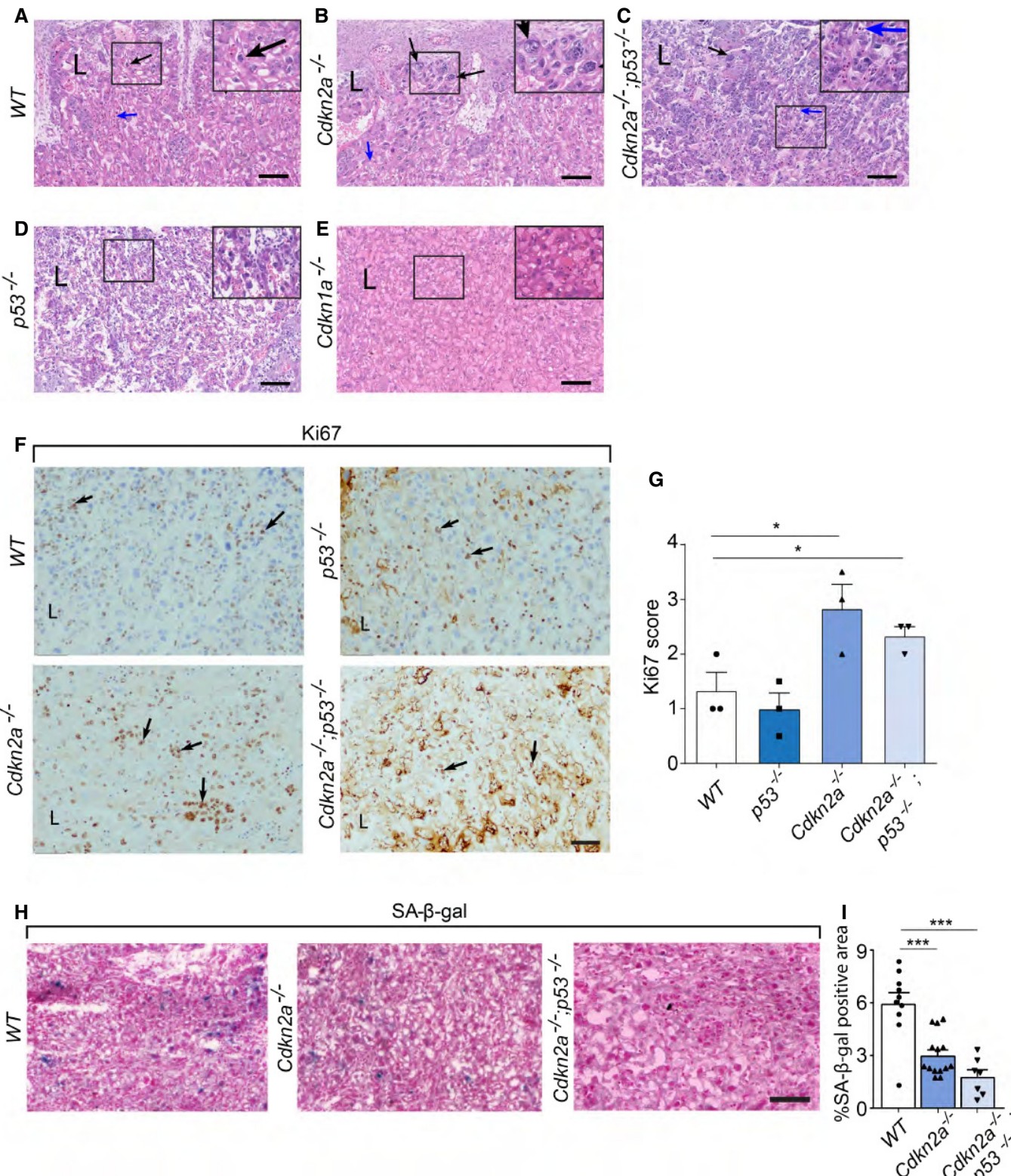

**Figure 3.**

**Figure 3.  Murine placentas with attenuated senescence programs exhibit morphological anomalies in the labyrinth.**
Histological evaluation of H&E-stained sections in the labyrinth zone of murine $Cdkn2a^{-/-}$ ($n = 5$ placentas), $Cdkn2a^{-/-};p53^{-/-}$ ($n = 5$), $p53^{-/-}$ ($n = 4$), and $Cdkn1a^{-/-}$ ($n = 3$) placentas on day E14.5, compared with WT placentas ($n = 6$).

A–E   The labyrinth zone of the WT (A) differs from those of $Cdkn2a^{-/-}$ (B) and $Cdkn2a^{-/-};p53^{-/-}$ (C) placentas. Black arrows indicate regular-sized nuclei in the WT placenta and polyploidy in the $Cdkn2a^{-/-}$ and $Cdkn2a^{-/-};p53^{-/-}$ placentas. Blue arrows indicate normal vasculature in the WT placenta and collapsed vasculature in the $Cdkn2a^{-/-}$ and $Cdkn2a^{-/-};p53^{-/-}$ placentas. Labyrinths of $p53^{-/-}$ (D) and $Cdkn1a^{-/-}$ (E) placentas did not differ from that of the WT placenta. L, labyrinth. Staining for each placenta was performed in triplicates.

F, G   Ki67 immunostaining in the labyrinth zones of murine WT, $p53^{-/-}$, $Cdkn2a^{-/-}$, and $Cdkn2a^{-/-};p53^{-/-}$ placentas, on day E14.5 ($n = 3$ placentas from each genotype). (F) Representative images of Ki67 staining in the labyrinth zone are shown. Positive Ki67 trophoblast cells are indicated with arrows. L, labyrinth. (G) Proliferation scores ($n = 3$ scores, derived from three placentas of each genotype) based on Ki67 staining show enhanced Ki67 expression in the placental labyrinths of $Cdkn2a^{-/-}$ and $Cdkn2a^{-/-};p53^{-/-}$ placentas relative to WT. Values are means + SEM of three scores for each genotype. *$P < 0.05$ by a one-tailed unpaired Student's $t$-test.

H   Representative images of SA-β-gal staining in the labyrinth zones of WT, $Cdkn2a^{-/-}$, and $Cdkn2a^{-/-};p53^{-/-}$ placentas on day E14.5.

I   SA-β-gal activity, quantified by ImageJ software in $Cdkn2a^{-/-}$ and $Cdkn2a^{-/-};p53^{-/-}$ labyrinths, is significantly reduced relative to those in WT labyrinth ($n \geq 7$ fields of view from each genotype). Values are means + SEM. ***$P < 0.001$ by a two-tailed unpaired Student's $t$-test.

Data information: Scale bars, 50 µm.

identified by βhCG secretion, had not yet developed (Fig 4A–C). In trophoblast cultures on day 5, gene set enrichment analysis (GSEA) revealed that pathways associated with pregnancy are significantly upregulated, as expected in these cells (Fig 4D and Appendix Table S4). Genes associated with organismal aging were also significantly enriched (Fig 4D). On the other hand, pathways associated with cell cycle, E2F targets, regulation of mitosis, and replication were downregulated (Fig 4D). Quantitative RT–PCR confirmed the increase in expression of the senescence-associated cell-cycle inhibitors *p16* and *p21*, as well as of *CCNE1*, yet another gene previously associated with senescence (Dulic *et al*, 1993; Fig 4E). Besides the increase in βhCG secretion (Fig 4C), syncytiotrophoblast differentiation was also confirmed by the increased expression of the trophoblast differentiation markers "glial cells missing" homolog 1 (GCM1), chorionic gonadotrophin-α (CGα), and βhCG itself on day 5 (Fig 4F). Therefore, the gene-expression profiles of our human placental cultures reflect cell-cycle arrest and trophoblast differentiation.

The SASP, composed of pro-inflammatory cytokines, chemokines, growth factors, and proteases, can modulate the senescent cell's microenvironment and promote its interaction with the immune system (Coppe *et al*, 2008; Lujambio *et al*, 2013; Sagiv & Krizhanovsky, 2013; Sagiv *et al*, 2016). We found that compared to 2-day cultures, our trophoblast cultures on day 5 indeed exhibited significantly upregulated gene sets associated with the immune response, cytokine activity, and major signaling pathways regulating the SASP, including JAK-STAT and MAPK (Fig 4D). We also observed enrichment of genes with promoter regions containing motifs of the transcription factor NF-κB (Fig 4D), a major regulator of SASP (Acosta *et al*, 2008; Kuilman *et al*, 2008). Immunoblot analysis confirmed the upregulation of components of MAPK, NF-κB, and JAK-STAT regulatory pathways (p-MEK 1/2, p-p65, and p-STAT3), respectively, on 5-day cultures compared with cultures on day 2 (Fig 4G). On day 5, syncytiotrophoblast cells also showed a marked increase in expression of the angiogenic factor PDGF-C and in a number of SASP-associated cytokines, including CCL2, CCL5, CCL8, IL6, IL1β, and TGF-β (Fig 4H). One of these cytokines, CCL5, can attract non-cytotoxic decidual natural killer (dNK) cells, which are essential for maintenance of the placenta (Hanna *et al*, 2006; Rajagopalan & Long, 2012). Additionally, an increase in the expression of other SASP components—matrix metalloproteases

(MMPs), in particular MMP2, MMP3, and MMP9—was seen in trophoblast cultures on day 5 (Fig 4D and I). TGF-β is known to regulate the expression of MMPs via Smad and non-Smad pathways in tumor stromal cells (Krstic & Santibanez, 2014); TGF-β and p-Smad3 were also elevated in syncytiotrophoblast on day 5 (Fig 4G and H), suggesting that they play a role in regulation of MMP expression in the placenta. Therefore, syncytiotrophoblast cultures expressed secretory components and activated pathways regulating SASP. Overall, these results demonstrated that syncytiotrophoblast cultures display characteristics of both cellular senescence and the activation of molecular mechanisms essential for normal placental function.

### Senescence pathways regulate expression of gelatinases

Among the SASP components that we found to be strongly upregulated in the syncytiotrophoblast were MMP2 and MMP9 (Fig 4I). These MMPs, designated gelatinases on the basis of their enzymatic substrate, play a role in successful trophoblast invasion in early pregnancy (Bischof *et al*, 2003; Isaka *et al*, 2003; Staun-Ram *et al*, 2004; Hua *et al*, 2011). We therefore proceeded to examine how the abolishment of senescence pathways would affect the activity of MMP2 and MMP9 in murine placentas of the WT and the $Cdkn2a^{-/-};p53^{-/-}$ genotypes, as well as in human placenta complicated by IUGR. We compared the expression of senescence-related proteins by immunoblot analysis on E14.5 of two independent age-matched placentas of WT and of the $Cdkn2a^{-/-};p53^{-/-}$ genotype. We found that the $Cdkn2a^{-/-};p53^{-/-}$ placentas downregulated the expression of key components of pathways regulating SASP, including p-Mek1/2, p-p65, and p-Stat3 (Fig 5A). In addition, the levels of p-Smad3, a signaling molecule of the TGF-β pathway implicated in the regulation of gelatinase activity, were lower in the $Cdkn2a^{-/-};p53^{-/-}$ than in the WT placentas (Fig 5A).

We then examined the expression of Mmp2 and Mmp9 and gelatinase activity in the murine placenta. Quantitative RT–PCR showed that compared to the WT, there was a marked reduction in the expression of Mmp2 and Mmp9 in the $Cdkn2a^{-/-};p53^{-/-}$ placentas (Fig 5B and C). Owing that Mmp2 and Mmp9 are gelatinases, we assessed the gelatinase activity by *in situ* gelatin zymography in the labyrinth zones of the murine WT and $Cdkn2a^{-/-};p53^{-/-}$ placentas. We found that the gelatinase activity in these

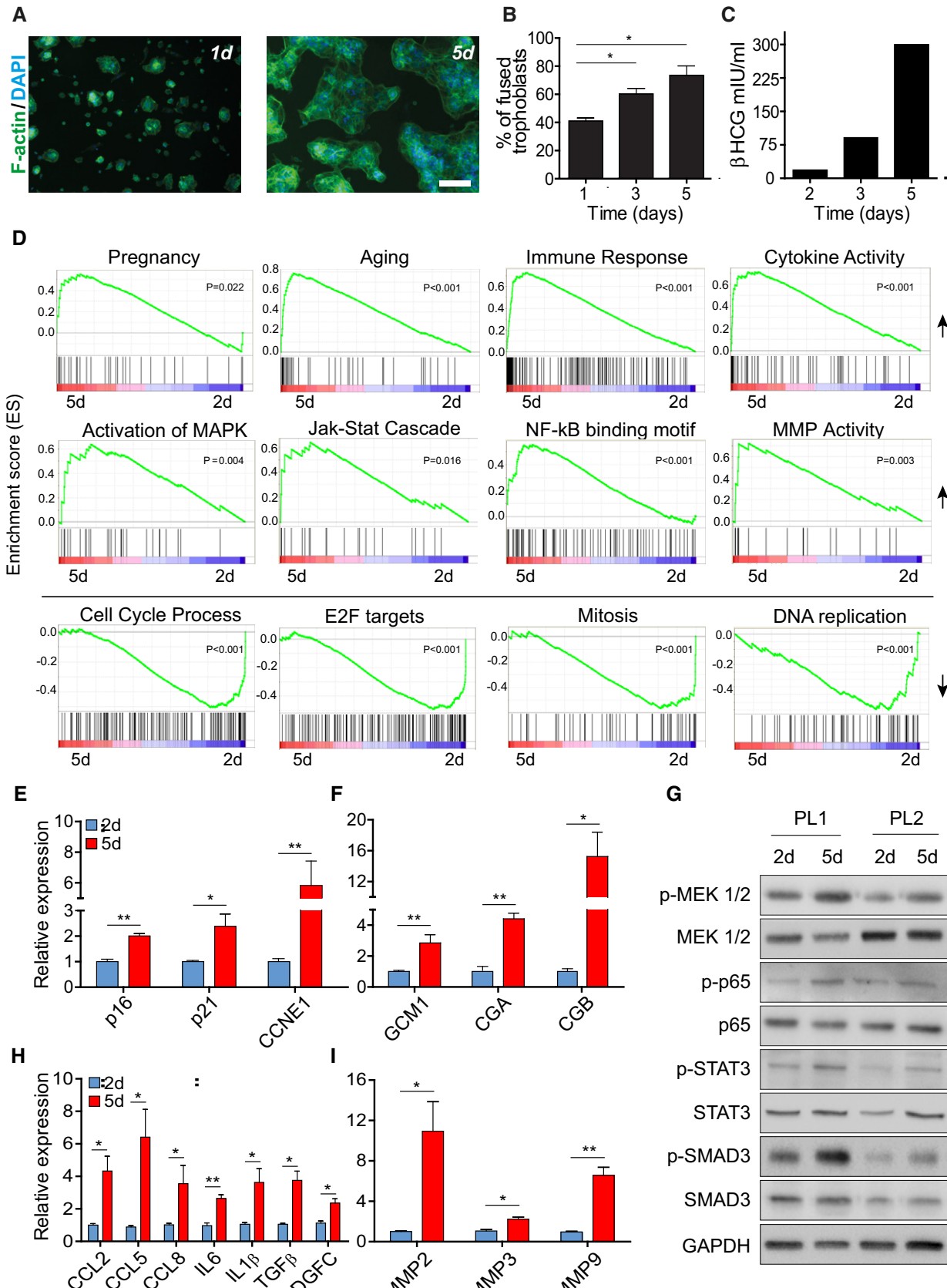

**Figure 4.**

◀

**Figure 4. Human primary syncytiotrophoblasts activate regulatory pathways of senescence.**

Human term placentas were dissected, and cytotrophoblast cells were extracted and seeded.

A, B  Cell fusion was monitored by staining for phalloidin (F-actin) (green) and DAPI (blue), on days 1 and 5 after seeding. Scale bar, 50 μm. (B) Trophoblast cell fusion was quantified on days 1, 3, and 5 post-seeding. Values are means + SEM, derived independently from human placentas of three pregnancies.

C  Quantification of the human βhCG hormone, normally secreted by functional syncytiotrophoblast cells, in the medium of a representative trophoblast culture on post-delivery days 2, 3, and 5.

D  Gene set enrichment analysis (GSEA) of published datasets from human primary trophoblast cultures on days 5 and 2 (n = 4 trophoblast culture of day 5 and day 2). Gene sets related to pregnancy, aging, immune response, cytokine activity, MAPK, JAK-STAT, NF-κB, and metalloproteinase activity were found to be enriched in day 5 cultures, whereas on the same day gene sets related to cell cycle, E2F targets, mitosis, and DNA replication were downregulated. Statistical significance was determined by permutation testing with normalized enrichment score (NES).

E, F  Quantitative RT–PCR analysis of expression of the cell-cycle-related genes p16, p21, and CCNE1 (E) and of the syncytiotrophoblast markers GCM1, CGA, and CGB (F).

G  Immunoblot analysis of the indicated proteins in human trophoblast cells, derived independently from two placentas, on days 2 and 5.

H, I  Quantitative RT–PCR analysis of expression of the SASP components CCL2, CCL5, CCL8, IL6, IL1β, TGF-β, and PDG-FC (H), and of the metalloproteases MMP2, MMP3, and MMP9 (I), in human primary trophoblast cultures on days 5 and 2.

Data information: Quantitative RT–PCR values are expressed as means + SEM. Results were obtained from trophoblast cultures derived from human placentas of four independent pregnancies. All RT–PCR experiments were performed in triplicates and repeated at least three times. Statistical significance was determined by unpaired two-tailed Student's t-test *P < 0.05; **P < 0.01.

zones was significantly reduced in the $Cdkn2a^{-/-};p53^{-/-}$ placentas compared with WT (Fig 5D and E). These results showed that gelatinase expression and activity are downregulated in murine placentas with disrupted senescence pathways.

The syncytiotrophoblast of the human placenta complicated by IUGR showed a significant reduction in the expression of senescence-related genes (Fig 1). We compared the gelatinase activity and expression in normal and IUGR-affected placentas. Using *in situ* gelatin zymography, we found that gelatinase activity was significantly reduced in the IUGR placenta (P = 0.008; Fig 5F and G). Having also found that SASP components and their regulatory pathways NF-κB and JAK-STAT were upregulated in human syncytiotrophoblast cultures (Fig 4D, G and H) and downregulated in murine placentas lacking p53 and Cdkn2a (Fig 5A), we compared the activation levels of these pathways in human normal and IUGR placentas. Immunoblot analysis revealed that the expression of p-p65 and p-STAT3 proteins (from the NF-κB and JAK-STAT pathways, respectively) was reduced in IUGR placentas relative to the normal ones (Fig 5H). In line with the observed reduction in gelatinase activity, quantitative RT–PCR also revealed a significant reduction in MMP2 and MMP9 expression in IUGR placentas relative to normal ones (Fig 5I). Expression of the SASP cytokine IL6 and of the members and downstream targets of the JAK-STAT pathway (IFNAR1 and IFNAR2) were also downregulated in IUGR relative to normal placentas (Fig 5I). Altogether, we found that placentas complicated by IUGR exhibit a decrease in the expression of senescence mediators and a reduction in gelatinase activity, similar to the effects observed in the placentas of senescence-attenuated mice. We conclude that in both human and murine placenta, gelatinase expression and activity are regulated by molecular pathways associated with senescence and these pathways are attenuated in IUGR.

# Discussion

Cellular senescence has been implicated in pathological processes in the adult organism, playing crucial roles in tumorigenesis, tissue repair, and aging. Here, we showed that molecular mechanisms of senescence maintain placental structure and function. We further showed that abolishment of the senescence program occurs in the human placental pathology of IUGR and that it leads to functional and morphological abnormalities in the murine placenta. Remarkably, the same regulatory pathways and matrix remodeling gelatinases are compromised in murine senescence-attenuated placentas and in the human placental pathology of IUGR, indicating an important role for these molecular machineries in maintaining placental integrity. The reduction in gelatinase activity that results from downregulation of senescence pathways may ultimately promote placental dysfunction and the onset and progression of IUGR, suggesting that mechanisms of senescence might play a critical role in human pregnancy.

Cellular senescence that occurs during placental development might derive from the same evolutionary origin as the damage-induced senescence caused by fusogens and viral infections (Chuprin et al, 2013; Goldman-Wohl & Yagel, 2014; Cox & Redman, 2017). One such fusogen of ancient retroviral origin, ERVWE1, mediates the establishment and expansion of the maternal–fetal multinuclear syncytiotrophoblast layer of the placenta and also serves as a trigger for the induction of cellular senescence (Chuprin et al, 2013). The syncytiotrophoblast of the normal human placenta exhibits features characteristic of senescent cells, including SA-β-gal activity, expression of senescence markers p53, DCR2, and the cyclin-dependent kinase (CDK) inhibitors p16 and p21, and the lack of proliferation (Chuprin et al, 2013). Thus, cell-fusion-induced senescence of syncytiotrophoblast leads to the activation of senescence-related pathways, similar to those activated during oncogene- and damage-induced senescence. Since the process of fusion-induced differentiation of cytotrophoblast into syncytiotrophoblast occurs at the early stages of placental formation and throughout pregnancy, we propose that senescence-related pathways might be activated at early stages of embryonic development during the first trimester of pregnancy. Nevertheless, disruption of this process may lead to manifestations only at a later stage of pregnancy. Indeed, the lack of cell fusion in syncytin knockout mice leads to changes in placental structure and function only starting from E14.5 (Dupressoir et al, 2009, 2011), while fusion itself starts at an early stage. Similarly, in human placental pathology of IUGR, defects in the molecular pathways might exist already early during embryonic development, but lead to onset of IUGR later during development due to increased functional load in the placenta.

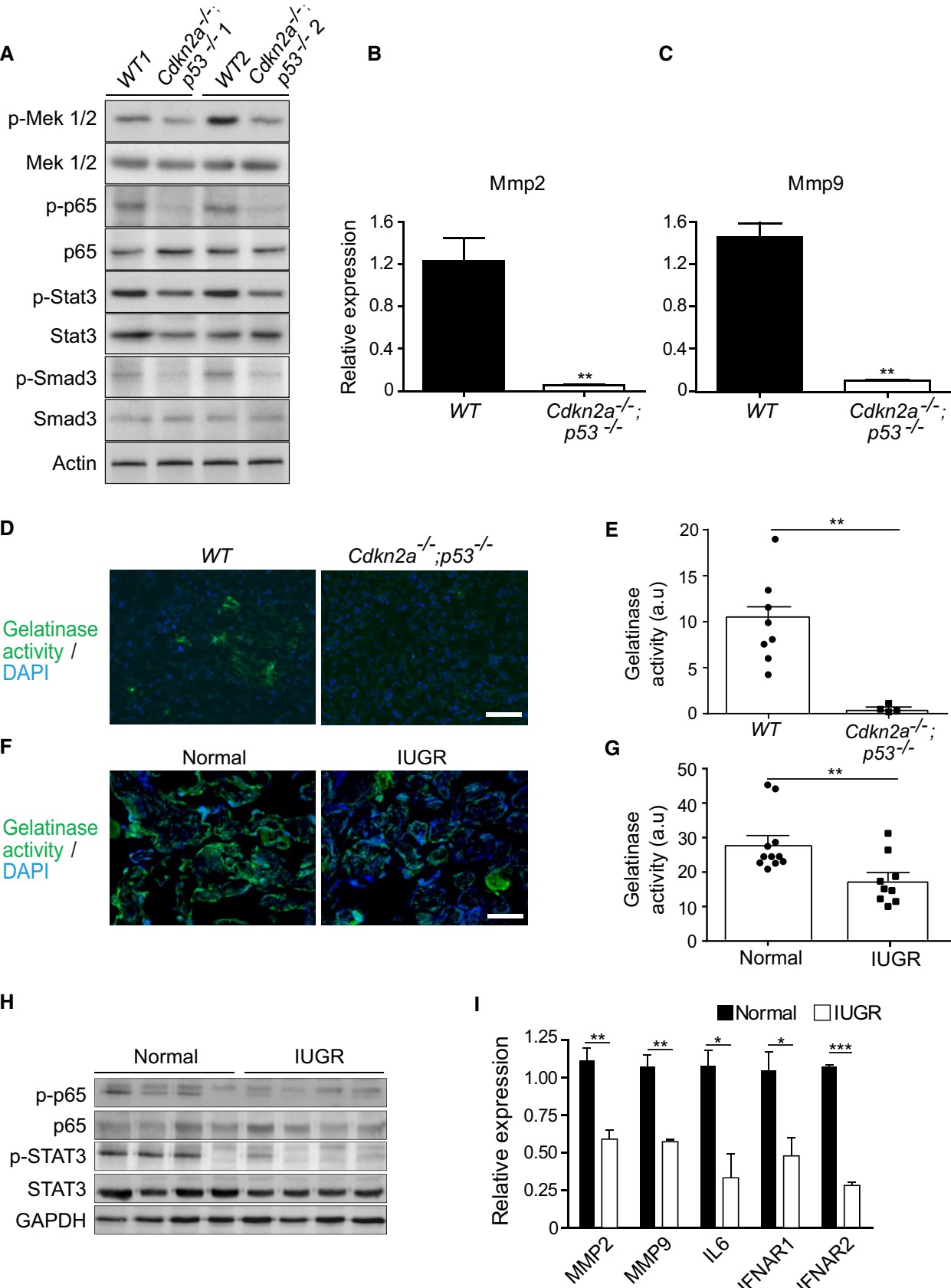

Figure 5.

◀

**Figure 5. Senescence pathways regulate gelatinase expression.**

A    Immunoblot analysis of the indicated proteins in WT and $Cdkn2a^{-/-};p53^{-/-}$ placentas, on day E14.5 (two independent placentas out of four from each genotype are shown).

B, C    RT–PCR analysis of expression of *Mmp2* and *Mmp9* in murine $Cdkn2a^{-/-};p53^{-/-}$ and in WT placentas on day E14.5 ($n \geq 4$ fields of view for each genotype). Values are means + SEM of at least three experimental repeats.

D, E    Representative images (D) and quantification (E) of *in situ* gelatin zymography of the labyrinth zone of murine *WT* and $Cdkn2a^{-/-};p53^{-/-}$ placentas on day E14.5. Scale bar, 50 μm.

F, G    Representative images (F) and quantification (G) of *in situ* zymography of human normal third-trimester post-partum placentas and placentas from IUGR-complicated pregnancies of the same gestational age ($n \geq 9$ fields of view of normal and IUGR placentas). Scale bar, 100 μm.

H    Immunoblot analysis of the protein content of p-p65, p65, p-STAT3, and STAT3 in normal placentas ($n = 4$) and placentas from IUGR-complicated pregnancies ($n = 4$). Each lane represents one independent placenta.

I    Quantitative RT–PCR analysis of expression of *MMP2*, *MMP9*, *IL6*, and the JAK-STAT targets *IFNAR1* and *IFNAR2*, in normal placentas and placentas from IUGR-complicated pregnancies. Results are from four human normal placentas and four placentas from IUGR-complicated pregnancies.

Data information: Values are means + SEM of at least three experimental repeats. Statistical significance was determined by unpaired two-tailed Student's *t*-test *$P < 0.05$; **$P < 0.01$; ***$P < 0.001$.

Senescence associated with embryonic development occurs not only in the placenta but also in transient fetal structures (Munoz-Espin *et al*, 2013; Storer *et al*, 2013). Both embryonic and placental senescence share features of SA-β-gal activity and proliferation arrest. However, whereas cell fusion is the main trigger of senescence in the placenta, it was suggested that senescence in the embryo is triggered by developmental cues and is aimed at tissue remodeling and organ patterning in the early embryo. Furthermore, while senescence in the placenta is dependent on the activation *of Cdkn2a* and *p53*, senescence in the developing embryo does not appear to involve these genes, nor is it dependent on the activation of a DNA damage response. Instead, senescence in the embryo is mediated mainly by p21 and is regulated by the TGF-β/SMAD and FOXO/PI3K signaling pathways (Munoz-Espin *et al*, 2013; Storer *et al*, 2013). These p21-expressing cells are able to re-enter the cell cycle and contribute to the adult tissues (Li *et al*, 2018), while senescent cells in the placenta are terminally arrested and lost during delivery. Therefore, although senescence in the developing embryo and in the placenta shares some common features, they utilize different molecular mechanisms for its induction and regulation. Of note, the use of complete knockout mice in these studies does not allow to exclude any indirect effects of these knockouts on different aspects of embryonic development, including the placenta.

The human placenta differs from mouse placenta in its morphogenesis, although they share some common features. Both placentas are hemochorial, meaning that the fetal trophoblast tissue is directly bathed in maternal blood. The human placenta is composed of one syncytial layer, formed during the first trimester of pregnancy and controlled by the ERWVE1 fusogen in the chorionic villi. In these villi, cytotrophoblastic cells continuously differentiate by cell fusion throughout pregnancy to generate the syncytiotrophoblast layer. The mouse placenta reaches maturity by E14.5, whereby trophoblast cells cease to proliferate. The mouse placenta consists of three layers: the labyrinth, the spongiotrophoblast, and the maternal decidua. The labyrinth compartment contains two syncytial layers, ST-l and ST-ll, whose formation is controlled by endogenous retroviral genes syncytin A and syncytin B (Dupressoir *et al*, 2009, 2011).

Why the induction of cellular senescence is required for syncytiotrophoblast formation and expansion remains to be elucidated. One possible reason is that the resistance of senescent cells to apoptosis (Wang *et al*, 1994; Yosef *et al*, 2016) maintains syncytiotrophoblast viability throughout pregnancy. This resistance might be attributed to the upregulation of anti-apoptotic proteins of the BCL-2 protein family, known to maintain the viability of senescent cells in other cell types (Yosef *et al*, 2016, 2017). Interestingly, the occurrence of apoptosis is diminished in syncytiotrophoblast cells after the 7[th] week of human pregnancy, and an increase in BCL-2 expression is observed in these cells as the pregnancy advances (Ishihara *et al*, 2000). Another possible reason has to do with the morphology of senescent cells, which are characterized by an increase in cell size (Campisi, 2011; Biran *et al*, 2017). Such a structure would facilitate the expansion of the multinuclear syncytiotrophoblast tissue (estimated at 13-fold between 12 weeks and term), providing the fetus with an increased transfer area (Georgiades *et al*, 2002; Goldman-Wohl & Yagel, 2014; Cox & Redman, 2017). Thus, in addition to the arrest of terminal proliferation and characteristic morphological changes, senescence might sustain syncytiotrophoblast viability and support the maternal–fetal interface expansion and nutrient transfer.

Secretion of SASP components, including cytokines, chemokines, and matrix remodeling enzymes, is a characteristic of senescent cells. Besides the protective cell-autonomous effects of senescence pathways in the syncytiotrophoblast, this secretion may also function to maintain fetal–maternal homeostasis (Bowen *et al*, 2002; Zhu *et al*, 2012). One SASP component, the chemokine CCL5, is highly expressed in both human primary trophoblasts and fibroblasts transduced with ERVWE1 (Chuprin *et al*, 2013). CCL5 is involved in the attraction of dNK cells to the maternal–fetal interface, regulating placental developmental processes, including trophoblast invasion and vascular growth (Hanna *et al*, 2006). Thus, molecular pathways of senescence might regulate the interaction of trophoblasts with the immune system and contribute to placental homeostasis.

MMPs, including MMP2 and MMP9, are expressed at the human feto-maternal interface by decidual cell populations, including trophoblasts (Anacker *et al*, 2011). Regulation of their activity at the maternal–fetal interface is critical for successful implantation and placentation (Alexander *et al*, 1996; Cohen *et al*, 2006; Zhu *et al*, 2012). The MMP2 and MMP9 gelatinases are intricately involved in placentation and are significantly reduced in senescence-attenuated mice and in the human placental pathology of IUGR. Interestingly, fetal single-nucleotide polymorphisms in the MMP2 and MMP9

genes are associated with increased risk of IUGR (Gremlich *et al*, 2007). Moreover, deficiency of MMP9 results in impaired reproduction and phenocopies aspects of preeclampsia and IUGR in mice (Plaks *et al*, 2013). Therefore, activation of molecular senescence-associated pathways in syncytiotrophoblast cells supports vital processes of placental formation and function in a cell non-autonomous manner.

IUGR is a major pathological consequence of the dysregulation of placental function (Regnault *et al*, 2002; Rizzo & Arduini, 2009; Veerbeek *et al*, 2014). Interestingly, oxidative and endoplasmic reticulum stresses in the placental trophoblast cells have been implicated in the pathophysiology of pregnancy complications, including IUGR (Burton & Jauniaux, 2004; Burton *et al*, 2017). We found here that the expression of senescence markers and the activation of regulatory pathways associated with senescence are significantly downregulated in human IUGR. This suggests that disruption of the senescence mechanisms may underlie IUGR pathogenesis. Consistently with the idea that senescence pathways contribute to syncytiotrophoblast viability, IUGR placentas show evidence of increased apoptosis, accompanied by a reduction in BCL-2 expression (Smith *et al*, 1997). Another possible consequence of the dysregulation of senescence pathways is a decrease in the attraction of dNK cells to the vicinity of the maternal–fetal interface, thus impairing maternal–fetal homeostasis and triggering the onset of IUGR. NK cells can be activated by NKG2D, which is involved in the interaction of senescent cells with NK cells (Sagiv *et al*, 2016). Placental bed biopsies from cases of IUGR indeed show a reduction in dNK cells (Williams *et al*, 2009). In addition, IUGR placentas exhibit increased signs of impaired telomere homeostasis (Davy *et al*, 2009; Biron-Shental *et al*, 2010, 2016). It is possible that the cytotrophoblast cell population undergo increased proliferation as a compensatory mechanism to maintain the integrity of compromised syncytiotrophoblast. Such increased proliferation leads to telomere shortening in the cytotrophoblast population. Therefore, deregulation of senescence mechanisms in the placenta may contribute to the onset and progression of the placental pathology of IUGR at multiple cellular levels.

We propose that cell-fusion-induced senescence of syncytiotrophoblast is an essential mechanism mediating the development and normal functioning of the placenta during pregnancy. The placental syncytiotrophoblast activates the main regulatory pathways of senescence, the same pathways as those activated in oncogene- or damage-induced senescence. Activation of these pathways contributes to the integrity and functioning of the syncytiotrophoblast in a cell-autonomous manner by sustaining its non-proliferative state and protecting its viability, and in a cell non-autonomous manner via SASP components that regulate its interaction with the immune system and tissue remodeling. Therefore, dysregulation of senescence pathways leads to placental abnormalities in mice and might provide a molecular explanation for some cases of IUGR in humans. Although further studies are needed, these results might offer an initial link between the activation of molecular senescence pathways during embryonic development and a pathological condition in humans. In suggesting a novel molecular explanation for pathologies associated with placental insufficiency, this study might help toward development of strategies for their prevention and treatment.

# Materials and Methods

## Animals

The following genetically modified mice were used: *Cdkn1a* null (Deng *et al*, 1995), *p53* null (Jacks *et al*, 1994), *Cdkn2a* null (Serrano *et al*, 1996), and *Cdkn2a/p53* null. Genotyping protocols for all genotypes have been previously described (Krizhanovsky *et al*, 2008; Yosef *et al*, 2017). All of the genetically modified mice were analyzed in parallel to WT or HET littermates. For analyses of WT mice, C57BL/6 mice were used. Interbreeding of heterozygous parents was used for DCE-MRI studies. All experiments were approved by the Weizmann Institute's Institutional Animal Care and Use Committee.

## *In vivo* contrast-enhanced MRI studies

Pregnant WT mice (8–10 weeks old) or *Cdkn1a p53 Cdkn2a* and *Cdkn2a/p53* genotypes were analyzed by MRI on day E14.5. MRI experiments were performed on a 9.4 Tesla (BioSpec 94/20 USR system; Bruker, Germany) equipped with a gradient coil system capable of producing pulse gradients of up to 40 gauss/cm in each of the three directions. A quadrature volume coil with a 72 mm inner diameter and a homogeneous RF field of 100 mm along the axis of the magnetic field was used for radiofrequency pulses. During MRI scanning, mice were anesthetized with isoflurane (3% for induction, 1–2% for maintenance), mixed with oxygen (1 l/min), and delivered through a nasal mask. Once anesthetized, the mouse was placed in a supine position in a head holder to ensure reproducible positioning inside the magnet. Respiration rate was monitored and kept at around 30–45 breaths per minute throughout the experimental period. During imaging, body temperature was maintained using a circulating water system adjusted to keep the mouse body temperature at 37°C during data acquisition. The embryo positioning within the pregnant female during DCE-MRI scan was noted. Following DCE-MRI scan, pregnant females were euthanized, and embryos and placentas were extracted and matched to their original position. The embryonic genotypes were assessed by sampling from each embryo using standard PCR-based genotyping protocol used for the genotyping of the parents.

## Contrast preparation

Biotin-BSA-GdDTPA (Plaks *et al*, 2011b) was derived from bovine serum albumin (BSA) by conjugation with biotin and GdDTPA (biotin-BSA-GdDTPA; approximately 80 kDa; relaxivity of 177 mM$^{-1}$ s$^{-1}$ per albumin and 7.55 mM$^{-1}$ s$^{-1}$ per Gd at 4.7T). An i.v. bolus dose of biotin-BSA-GdDTPA (4 μmol/kg (Gd: 92 μmol/kg) in 200 μl of PBS) was injected during dynamic (macromolecular) contrast-enhanced MRI to allow evaluation of vascular function, including blood volume and vascular permeability. The biotin label was used to visualize the distribution of the contrast material in histological sections.

## DCE-MRI data acquisition

For dynamic contrast-enhanced MRI imaging, T$_1$-weighted 3D-GRE images were acquired from the time of biotin-BSA-GdDTPA

administration and up to 67 min 55 s (25 scans), using the following parameters: pulse flip angle = 15°; TR = 10 ms; TE = 3 ms; two averages; FOV = 4 × 4 × 4 cm; matrix = 256 × 256 × 64; and acquisition time = 2 min 43 s.

### DCE-MRI data analysis

Placental reactive oxygen intermediates were marked by Analyze Direct 11.0 (AnalyzeDirect, www.analyzedirect.com) software for calculation of the mean signal intensity (SI). For some of the placentas (in which there were fetal movements), the region of interest was manually marked for each scan. The rate of change in contrast material concentration was scaled to the concentration in the vena cava (ROE = rate of enhancement with units of $min^{-1}$). To quantify the changes in SI dynamics, parameters of initial enhancement and recovery were defined. SI initial enhancement was defined as the ratio between the SI in the first local maximum and the initial SI. SI recovery was defined as the ratio between the SI in the second local maximum and the SI in the local minimum point between the two maximum points (roughly at the middle of the measurement time). Differences in the initial enhancement and recovery parameters in the null placentas from each genotype were compared with the global WT placentas by one-way ANOVA followed by Tukey's *post hoc* test.

### Histopathology of mouse placenta

Paraffin-embedded murine placental tissues (day E14.5) were serially sectioned (4 μm thickness) and stained with hematoxylin and eosin for histopathological evaluation. At least three null placentas derived from each genotype and three WT or HET placentas from their respective littermates were stained and evaluated by an experienced certified pathologist (A.B.).

Immunostaining was performed using the Ki67 antibody on placental sections of *WT, p53$^{-/-}$, Cdkn2a$^{-/-}$, and Cdkn2a$^{-/-}$; p53$^{-/-}$* (at least three placentas from each genotype). The frequency of Ki67-positive cells was semi-quantitatively scored in an unbiased manner, on a scale of 0–4 (at 0.5 intervals), where 0 = negative staining, 0.5 = < 5% of trophoblasts showing positive staining, 1 = 5–10%, 1.5 = 10–25%, 2 = 25–33%, 2.5 = 33–50%, 3 = 50–66%, 3.5 = 66–75%, and 4 = more than 75% of trophoblasts showing Ki67 positivity. Only cells of trophoblastic lineage were scored and other cellular lineages (e.g., vascular endothelium, decidua, and blood cells) were omitted from the scoring.

SA-β-gal activity was detected as described previously (Krizhanovsky *et al*, 2008) and quantified by ImageJ software (NIH).

### Isolation of human primary cytotrophoblast

The study, including all the experiments with human samples as described below in this and later paragraphs, was approved by the Institutional Review Board of Meir Medical Center, and Helsinki Committee Approval was received by T.B.S. Placental villous tissues (30 g) were isolated from human term placentas after delivery by an established method (Kliman *et al*, 1986). Briefly, isolated villous tissue was weighed, washed three times with PBS at room temperature, and submitted to sequential trypsin-DNase digestions at 0.25% as previously described (Kliman *et al*, 1986). Cell suspensions were

carefully layered over a discontinuous Percoll gradient (70–5%, in 5% steps) and centrifuged. The middle layer, representing a highly enriched population of viable mononuclear trophoblastic cells, was removed and washed with DMEM. Cells were diluted to $0.5 \times 10^6$/ml with DMEM, supplemented with 2 mm glutamine, 10% heat-inactivated fetal calf serum, 25 mm HEPES, 100 IU/ml penicillin, and 100 μg/ml streptomycin, then plated in 6-well plates ($1 \times 10^6$ cells in 2 ml/well), and incubated at low oxygen (5%), 37°C culture. Cells obtained by the Percoll procedure were examined after 1.5, 2, 3, and 5 days in culture by immunofluorescence staining and gene-expression analysis. Immunofluorescence analysis of trophoblast cells was performed with trophoblast markers α-syncytin, α-cytokeratin 7 (Santa Cruz Biotechnology), and DAPI (Sigma).

### Human tissue collection and analysis

Immediately after delivery, placental biopsies were obtained under sterile conditions from four women whose pregnancies were complicated by IUGR and from four uncomplicated placentas as controls. Biopsies were taken from the intermediate trophoblast area, midway between cord insertion and the edge of the placenta. The biopsied material (~ 1 cm³) was either fixed in formalin and embedded in paraffin or stored at −80°C for evaluation by immunoblotting. Paraffin-embedded placental sections of 4 μm were deparaffinized and rehydrated in a series of ethanol solutions of decreasing concentration. Antigen retrieval was performed in a hot water bath, and sections were blocked for non-specific binding with 4% horse serum and 1% BSA. Primary antibodies recognizing p15 (Assay Biotech), Ki67, p16 (Abcam), p21 (BD Biosciences), and p53 (Santa Cruz Biotechnology) were applied overnight at 4°C. Detailed information about the antibodies is available in the supplementary (Appendix Table S5). Staining was developed using a DAB substrate kit (Vector Laboratories) followed by hematoxylin counterstaining. Sections were visualized with an Olympus microscope, and images were analyzed with CellP software (Diagnostic Instruments). The percentage of positive syncytia was presented as the ratio of syncytiotrophoblast cells containing one or more DAB-positive nuclei (denoted 0 or 1) out of all counted syncytiotrophoblast per field. The average number of positive syncytia out of all syncytia in a particular field of view was calculated as a mean of at least 12 counting fields, derived from three normal or IUGR placental sections, in a systematic, random manner, using the ImageJ software.

### Immunoblotting

Human and mouse placental tissues and human cultured trophoblast cells were lysed in Ripa buffer (50 mM Tris–HCl pH 7.4, 150 mM NaCl, 1 mM EDTA, 1% Na-D.O.C., 0.1% SDS, 1% Triton X-100) containing protease inhibitors (Sigma), using a tissue homogenizer. Equal amounts of protein were separated on 12% SDS–polyacrylamide gels and transferred to PVDF membranes. Detection was performed with the following primary antibodies: α-p21 (BD Biosciences), α-p53, α-Stat3, α-p65 (Santa Cruz Biotechnology), α-p16 (Abcam), α-DCR2 (Enzo Life Sciences), α-Smad3, α-p-Smad3, α-p-Stat3, α-p-p65, α-Mek1/2, α-p-Mek1/2 (Cell Signaling), α-β-actin (Sigma), and α-GAPDH (Merck). The detailed list of antibodies is presented in Appendix Table S5.

**Expression array analysis and quantitative RT–PCR**

Total RNA was extracted from human primary trophoblast cultures at days 1.5, 2, 3, and 5 after seeding, using an RNeasy Mini Kit (Qiagen). RNA purity was assessed with a NanoDrop (ND-1000) Spectrophotometer (Peqlab Biotechnologie) and the BioAnalyzer 2100 system (Agilent Technologies). cDNA was prepared, labeled, and hybridized to GeneChip PrimeView Human Gene Expression Array (Affymetrix) according to the manufacturer's protocols. Hybridized chips were stained, washed, and scanned with the Affymetrix Scanner GeneChip 3000 7G Plus and converted to cell-intensity files by Affymetrix Expression Console Software. Preprocessing was performed with the Robust Microarray Averaging algorithm (Irizarry *et al*, 2003). Gene set enrichment analysis (GSEA, http://broadinstitute.org/gsea) was applied using annotations from KEGG (Subramanian *et al*, 2005) to determine whether predefined gene sets showed enrichment in late trophoblast cultures (day 5). Gene sets showing $P < 0.05$ between day 5 and day 2 cultures were considered enriched. The list of enriched GSEA groups with their related links is presented in Appendix Table S4. The list of all gene members of each group is available (Dataset EV1). Quantitative RT–PCR was performed on human cultured trophoblast cells. Total RNA was converted into cDNA using M-MMLV reverse transcriptase (Promega). Quantitative real-time PCR was performed with an ABI PRISM 7700 (Life Technologies) Sequence Detection System, using Platinum SYBR Green qPCR Supermix (Invitrogen), as described previously (Yosef *et al*, 2017). Primer sequences are available on request.

**_In situ_ zymography**

Human and mouse placental-frozen sections (10 μm) were washed with developing buffer containing 150 mM NaCl, 5 mM $CaCl_2$, 100 mM Tris–HCl pH 7.6, 20 μM $ZnCl_2$, and 0.05% Brij-35. DQ-gelatin (E-12054; Molecular Probes) was diluted to a concentration of 20 μg/ml with developing buffer. DQ-gelatin solution was applied on each section (100 μl), and slides were incubated at 37°C for 4 h in a dark wet chamber. Following incubation, slides were washed three times with TBS and 150 nM NaCl and then fixed with 4% paraformaldehyde for 10 min. Sections were washed three times for 1 min with PBS + DAPI. Gelatinolytic activity was observed by fluorescence microscopy (Olympus) as green fluorescence (absorption maxima, ~ 495 nm; fluorescence emission maxima, ~ 515 nm). Gelatinase activity was quantified from images using ImageJ software.

**Statistical analysis**

All experiments were performed in at least three independent repeats. Two-tailed Student's *t*-test (unpaired) was used to analyze differences between groups, with $P < 0.05$ considered statistically significant. For MRI analysis, one-way ANOVA with *post hoc* Tukey's HSD was applied. Data are presented as means + SEM.

# Data availability

The microarray data from this publication have been deposited to the GEO database (https://www.ncbi.nlm.nih.gov/geo) and assigned an accession number GSE118351.

Expanded View for this article is available online.

## Acknowledgements

We thank S. Yagel and D. Goldman-Wohl (Hadassah Hospital, Jerusalem) for insightful suggestions, I. Sher for graphical help with the figures, I. Orr for helping to analyze the microarray data, and all members of Krizhanovsky laboratory for helpful discussions. This work was supported by grants to V.K. from the European Research Council under the European Union's FP7 (309688) and from the Israel Science Foundation (634-15). This work was supported by the Seventh Framework European Research Council Advanced Grant 232640-IMAGO and by National Institutes of Health (Grant 1R01HD086323-01) to M.N. Michal Neeman is the incumbent of the Helen and Morris Mauerberger Chair in Biological Sciences.

## Author contributions

HG, SS, and EV designed and performed the experiments. ML and MN designed, performed, and analyzed the DCE-MRI experiments. RR performed statistical analyses of the data, including DCE-MRI data. SAY and AB performed the histological experiments, scored and analyzed the immunohistochemical stainings, and contributed to data analysis. KT-G and TB-S obtained the human placental samples. HG and VK analyzed the experiments and wrote the article. VK supervised the project. All authors discussed the results and commented on the article.

## Conflict of interest

The authors declare that they have no conflict of interest.

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
