## [Review Process File · The EMBO Journal]

Molecular pathways of senescence regulate placental structure and function

Hilah Gal, Marina Lysenko, Sima Stroganov, Ezra Vadai, Sameh A.Youssef, Keren Tzadikévitch-Geffen, Ron Rotkopf, Tal Biron-Shental, Alain de Bruin, Michal Neeman and Valery Krizhanovsky

Review timeline:

Submission date:	4th Oct 2018
Editorial Decision:	19th Dec 2018
Revision received:	8th May 2019
Editorial Decision:	2nd Jul 2019
Revision received:	12th Jul 2019
Accepted:	26th Jul 2019

Editor: Deniz Senyilmaz Tiebe

Transaction Report:

1st Editorial Decision

19th Dec 2018

Thank you for submitting your manuscript for consideration by the EMBO Journal. I would like to apologize for the delay in getting back to you. It took longer than anticipated to receive the full set of referee reports due this being a very busy time of year.

Your manuscript has now been seen by three referees whose comments are shown below. As you can see, all referees express interest in the proposed link between senescence and placental structure/function. However, they also raise concerns that need to be addressed in full before we can consider publication of the manuscript here.

Given the referees' positive recommendations, I would like to invite you to submit a revised version of the manuscript, addressing the comments of all three reviewers. I should add that it is EMBO Journal policy to allow only a single round of revision, and acceptance of your manuscript will therefore depend on the completeness of your responses in this revised version. We require strong endorsement from our referees for publication here.

REFeree REPORTS:

Referee #1:

This work of Valery Krizhanovsky is the follow up of their previous publication (Churpin et al, Gene Dev, 2013), which revealed signs of senescence in the placenta, now asking the of whether the induction of cellular senescence is necessary for plzcenta. Through a series of elegant experiments, comparing the analysis of human placenta from pregnancies with intrauterine growth restriction (IUGR) and mouse placenta allowing functional and genetic studies, they provide evidence for a role of senescence in placenta structure and function. In agreement, a reduced level of senescence was observed in IUGR placenta. Overall, the experiments are convincely executed and presented.

Only a minor modification is requested in the Discussion section. The sentence " Since syncytiotrophoblasts do not proliferate, no telomere shortening can occur in these cells." is misleading since there are now examples of telomere shortening in non-dividing cells, see e.g. the work of Helen Blau: Chang et al, PNAS 2018.

Referee #2:

This manuscript is testing the hypothesis that senescence associated pathways are key regulators of syncytiotrophoblast development and establishment of placental exchange interphase, and that alterations in these pathways have an essential role in the pathogenesis of IUGR. This manuscript combines staining for senescence regulators in normal and IUGR human placentas, as well as knockout mouse models for senescence regulators. Authors propose that molecular mediators of senescence such as p16, p19 and p53 regulate placental structure and function and that IUGR is associated with reduced senescence characteristics in syncytiotrophoblasts. Using DCE-MRI, they show that concomitant deletion of p16 and 19 in mice leads to alteration in signal identity, implying compromised maternal circulation in the placentas. Using human term placental cytotrophoblast cultures to differentiate syncytiotrophoblasts, they show increased RNA expression for senescence regulators upon syncytiotrophoblast differentiation. Moreover, they show that IUGR placentas have reduced expression for p15,16 and 19 and p53, implying these pathways being compromised in IUGR. They also show severe reduction in MMPs and gelatinase activity, implying link to SASP and IUGR. While these findings are interesting, they are for the most part descriptive/correlative. The documentation of placental defects and expression of the senescence regulators is performed quite superficially, and it is not always clear which cell types are affected. More thorough assessment of the affected cell types would be critical to fully evaluate the findings. It would also be interesting to know if the hyperproliferative phenotype associated with knockout mouse placentas caused by persistence of proliferative labyrinth trophoblast progenitor cells. It is also not clear how well the mouse knockouts that lack these regulators in all cells, not just trophoblast lineage, can be used to make conclusions about trophoblast development specifically. Moreover, better description of the similarities and differences of mouse and human placentas would also be helpful. The developmental stage that is compared is also very different (E14.5 in mice would developmentally represent first trimester in human, but the IUGR placentas are collected at term). The potential value of this work is that it addresses the poorly understood placental pathophysiology in IUGR, and attempts to correlate mechanistic studies in mice to human. However, additional clarity to the data presentation and interpretation of the results is needed.

Specific comments:

1. The authors state "senescence might sustain SynT viability and support the fetal/maternal interface expansion and nutrient transfer". What is the basis for this statement? The concept of cellular senescence is typically associated to mitotic cells (see review Campisi et al, 2007). However, syncytiotrophoblasts are post-mitotic, multinucleated cells. Therefore, the premise of studying the role of senescence pathways in post-mitotic cells could be questioned. Are the authors implying that senescence is the mechanisms by which cytotrophoblasts become syncytiotrophoblasts? Please clarify. In the Results section, rephrase the statement "The contribution of senescence pathways to accurate placental function in humans is unknown" for more clarity of concept. It is very broad and vague statement.

2. In Figure 1 A-D, the immunostaining of nuclear markers of senescence using the DAB chromogen overlaps with the hematoxylin nuclear counterstain and is difficult to interpret. What was the criteria used to distinguish nuclei with DAB and hematoxylin co-stain from those with hematoxylin alone? As syncytiotrophoblasts are multinucleated cells, were there cells that contained a mixture of DAB positive and negative nuclei? Was the quantification of "% positive syncytia" based on ratio of DAB positive nuclei out of all nuclei or ratio of syncytiotrophoblast cells containing one or more DAB positive nuclei out of all counted syncytiotrophoblast cells? Were these regulators expressed in cytotrophoblasts or other cells in the placentas? Placentas from 4 uncomplicated and 4 IUGR pregnancies were assessed; but there are more data points in the quantification graph. Please clarify what each data point represents.

3. The authors used knockout mice for p16, 19 and 53, to investigate the hypothesis that the senescence associated pathways are required for proper placental development and function. However, these are complete knockouts, that lack these regulators in all tissues. It is therefore not clear how much indirect effects from other cell types contribute to the phenotype. Please address. Which other cell types in the mouse placentas express them?
4. The authors show that genetic ablation of factors associated with senescence is correlated with changes in the SI detected by DCE-MRI, which evaluates in vivo maternal circulation in the placenta. What about defects in fetal vasculature in the placenta? The characterization of the differences and similarities in trophoblast histology and placental vascularization between wild-type and mutants by H&E alone (Fig 3 A-E) is limited, and difficult to interpret. For instance, the authors report no significant changes in the Cdkn1a null placenta (Fig 3E); however, the placental architecture appears different from wild-type. More definition can be displayed by immunostaining for specific cell types. Please stain for trophoblast and endothelial specific markers to show better the placental structure and changes in labyrinth size, morphology etc. Also, it has been shown that syncytiotrophoblasts in the mouse placenta develop from proliferative Epcam+ labyrinth progenitor cells (LaTP, Ueno et al. Dev Cell 2013) that disappear between E12.5-14.5. If these knockout placentas show hyperproliferative phenotype, is this linked to persistent LaTP proliferation? The authors state increased hypercellularity in the labyrinth zones of Cdkn2a null and double Cdkn2a;p53 null placentas. How was cellularity assessed?
5. Also, The authors show an increase in cellular proliferation marker Ki67. It is not clear what cells is KI67 signal coming from? It would be helpful to show KI67 with additional placental markers for SynT, LaTP, vasculature etc. BRDU incorporation in the knockout placentas would also be helpful. Assessment of different developmental stages would also be helpful.
6. This manuscript lacks discussion of differences/similarities of human and mouse placentas, which would be important if results from knockout studies are extrapolated to humans. As an example, in mice, proliferative trophoblasts disappear by E 14.5, but in human, cytotrophoblasts continue to proliferate and differentiate to syncytiotrophoblasts. Also it would be important to discuss the developmental timing when the senescence pathways are proposed to influence syncytiotrophoblast development in mice and human.
7. IUGR is associated with placentas that are smaller and with altered chorionic villi architecture and cell composition. Do the authors propose that IUGR placentas are hyperproliferative? How does this result in smaller placentas?
8. Another group reported that p53 expression was predominantly present in cytotrophoblasts and increased instead of decreased in IUGR placentas (Levy et al, Amer J Ob Gyn, 2002). Can the authors comment on this discrepancy?
9. If possible, provide clinical characteristics of human pregnancies, namely, gestational age at delivery and birth weight, and state how IUGR was defined (including nomogram used to specify weight percentile).
10. Provide the references for the specific animal strains used. Given that some knockout strains are fertile, please clarify whether heterozygous or null mice were interbred to generate null and control (wild-type or heterozygous) littermates. If interbreeding of heterozygous parents was used, how were the embryonic genotypes assessed in order to match with DCE-MRI phenotypes?

Referee #3:

The authors previously showed that cell fusion induces senescence in syncytiotrophoblasts in placenta. In the present study, they show evidence that this senescence phenotype is important for placental structure and function, mainly using mouse genetics and human samples. This is highly

original and interesting study. Although the implication of senescence in placental function is correlative, the data seem solid and the study open a new research question.

Fig. 1: they first show a low frequency of senescence markers in IUGR placenta, raising an interesting possibility of functional link between syncytiotrophoblast senescence and placental function. In the IHC figs, the difference is unclear to me. e.g. p53 seems to be higher in IUGR? It would be helpful if they provide more detailed description of histology particularly for non-exerts.

Fig. 2B: In DCE-MRI, what does the 'recovery phase' correspond to? Again it would be helpful with some explanation for general audience.

Transcriptomic analysis: they define differentially expressed genes based on '2-fold change'. I wonder if they consider any statistics, e.g. *fdr*? Fig. 4d, they show GSEA plots. I understand they have used *msigdb*, but they should specify exactly which gene signatures they have used for each plot, and provide citation associated with them if applicable.

Do they see any phenotype in fetus / fertility in those mutant mice? This would be important to actually show a link between senescence (or p16/p63) and placental functions.

The implication of senescence deficiency in human IUGR is interesting (Fig. 1 and Fig. 5). Have they also checked proliferation in IUGR syncytiotrophoblasts in placenta? Also, I wonder if there is any evidence in the literature for developmental defects in KO mice for e.g. *gelatinases*, which might be due to placental issues? Although not definitive, these pieces of information would support the correlation between senescence/*sasp* and the placental function.

Minor points:

It would be advisable to provide more detailed information about antibodies used, catalogue numbers, dilutions.

Immunoblots in placenta samples: based on Methods, they use whole placenta lysates. Would this be sensitive enough to detect syncytiotrophoblasts-specific changes or do they see p16/p21/p53 expression in other parts of placenta as well (e.g. Fig. 1E)? Or do they somehow enrich syncytiotrophoblasts during sampling?

Please see next page.

Point by point response to the reviewers' comments

Molecular pathways of senescence regulate placental structure and function
Gal et al.

We thank the reviewers for their positive view on our work and for their helpful suggestions.

Referee #1 (Remarks to the Author):

This work of Valery Krizhanovsky is the follow up of their previous publication (Churpin et al, Gene Dev, 2013), which revealed signs of senescence in the placenta, now asking whether the induction of cellular senescence is necessary for placenta. Through a series of elegant experiments, comparing the analysis of human placenta from pregnancies with intrauterine growth restriction (IUGR) and mouse placenta allowing functional and genetic studies, they provide evidence for a role of senescence in placenta structure and function. In agreement, a reduced level of senescence was observed in IUGR placenta. Overall, the experiments are convincingly executed and presented.

Only a minor modification is requested in the Discussion section. The sentence " Since syncytiotrophoblasts do not proliferate, no telomere shortening can occur in these cells." is misleading since there are now examples of telomere shortening in non-dividing cells, see e.g. the work of Helen Blau: Chang et al, PNAS 2018.

We agree with the reviewer and removed the statement from the Discussion section of the manuscript.

Referee #2 (Remarks to the Author):

This manuscript is testing the hypothesis that senescence associated pathways are key regulators of syncytiotrophoblast development and establishment of placental exchange interphase, and that alterations in these pathways have an essential role in the pathogenesis of IUGR. This manuscript combines staining for senescence regulators in normal and IUGR human placentas, as well as knockout mouse models for senescence regulators. Authors propose that molecular mediators of senescence such as p16, p19 and p53 regulate placental structure and function and that IUGR is associated with reduced senescence characteristics in syncytiotrophoblasts. Using DCE-MRI, they show that concomitant deletion of p16 and 19 in mice leads to alteration in signal identity, implying compromised maternal circulation in the placentas. Using human term placental cytotrophoblast cultures to differentiate syncytiotrophoblasts, they show increased RNA expression for senescence regulators upon syncytiotrophoblast differentiation. Moreover, they show that IUGR placentas have reduced expression for p15,16 and 19 and p53, implying these pathways being compromised in IUGR. They also show severe reduction in MMPs and gelatinase activity, implying link to SASP and IUGR. While these findings are interesting, they are for the most part descriptive/correlative. The documentation of

placental defects and expression of the senescence regulators is performed quite superficially, and it is not always clear which cell types are affected. More thorough assessment of the affected cell types would be critical to fully evaluate the findings. It would also be interesting to know if the hyperproliferative phenotype associated with knockout mouse placentas caused by persistence of proliferative labyrinth trophoblast progenitor cells. It is also not clear how well the mouse knockouts that lack these regulators in all cells, not just trophoblast lineage, can be used to make conclusions about trophoblast development specifically. Moreover, better description of the similarities and differences of mouse and human placentas would also be helpful.

We addressed the above comments of the reviewer one by one as listed below in the specific comments.

The developmental stage that is compared is also very different (E14.5 in mice would developmentally represent first trimester in human, but the IUGR placentas are collected at term).

We are well aware of the differences in mouse and human placenta during embryonic development. We chose day E14.5 in mice since at this stage the mouse placenta has reached its mature form and previous studies revealed that deficiency in syncytiotrophoblast formation leads to placenta alterations at E14.5 and later during development (Dupressoir A. et al, PNAS, 2009, Dupressoir A. et al, PNAS, 2011). We refer to this point in the Results section under: "Molecular mechanisms of senescence affect placental DCE-MRI dynamics." Therefore, the developmental stage in mice was chosen to test the effect of the genotypes with impaired senescence function to placenta formation at the stage, when genetic differences that affect fusion can have an effect. This stage is also appropriate for DCE-MRI studies (Solomon E. et al, PNAS, 2014). This point is also mentioned in the Results of the manuscript, under the same section. In humans, normal and IUGR placentas were available to us upon delivery at term. This is the only material that is available for our study. The differences between mouse and human placenta are now discussed in details at the Discussion (Page 17).

The potential value of this work is that it addresses the poorly understood placental pathophysiology in IUGR, and attempts to correlate mechanistic studies in mice to human. However, additional clarity to the data presentation and interpretation of the results is needed.

Specific comments:

1. The authors state "senescence might sustain SynT viability and support the fetal/maternal interface expansion and nutrient transfer". What is the basis for this statement? The concept of cellular senescence is typically associated to mitotic cells (see

review Campisi et al, 2007). However, syncytiotrophoblasts are post-mitotic, multinucleated cells. Therefore, the premise of studying the role of senescence pathways in post-mitotic cells could be questioned. Are the authors implying that senescence is the mechanisms by which cytotrophoblasts become syncytiotrophoblasts? Please clarify.

In our previous study, we identified cell-cell fusion as a trigger of the senescence response in different cells, including trophoblasts (Chuprin A et al., Genes Dev, 2013). This fusion is mediated by the ERVWE1 fusogen, which is endogenously expressed in the human placenta. We propose that the natural process of differentiation of cytotrophoblast cells into syncytiotrophoblast by syncytial cytotrophoblast fusion, leads to the induction of senescence in syncytiotrophoblasts following fusion. Indeed, as the reviewer suggests, it appears that senescence is a mechanism by which syncytiotrophoblast develop their complete phenotype following cytotrophoblast fusion. We now further clarified this point in the text, under the introduction section of the manuscript: "ERVWE1, a fusogen of viral origin, is endogenously expressed in the placenta and mediates cell-fusion-induced senescence of the syncytiotrophoblast, following cytotrophoblast differentiation."

In the Results section, rephrase the statement "The contribution of senescence pathways to accurate placental function in humans is unknown" for more clarity of concept. It is very broad and vague statement.

The statement was rephrased as suggested to: "We aimed to understand the contribution of senescence associated pathways to placenta function".

2. In Figure 1 A-D, the immunostaining of nuclear markers of senescence using the DAB chromogen overlaps with the hematoxylin nuclear counterstain and is difficult to interpret. What was the criteria used to distinguish nuclei with DAB and hematoxylin co-stain from those with hematoxylin alone?

The criteria was based on image analysis by microscopy. The color of DAB staining is brown, while the color of Hematoxylin is violet and the colors can be distinguished in the microscope. We performed quantification of DAB positive (brown) cells based on the color. This is a standard procedure in immunohistopathology, owing the wide use of Eosin/Hematoxylin as a common counterstain for immunostainings.

As syncytiotrophoblasts are multinucleated cells, were there cells that contained a mixture of DAB positive and negative nuclei? Was the quantification of "% positive syncytia" based on ratio of DAB positive nuclei out of all nuclei or ratio of syncytiotrophoblast cells containing one or more DAB positive nuclei out of all counted syncytiotrophoblast cells?

Indeed, there were regions that contained a mixture of positive and negative cells. We counted each syncytial region as positive if it contained at least one positively stained nuclei. A detailed explanation of the quantification was added to "Materials and Methods" section under "Human Tissue Collection and Analysis".

Were these regulators expressed in cytotrophoblasts or other cells in the placentas?

In order to address this comment we expanded our analysis and performed immunofluorescence co-staining of senescence regulators p16 and p21, together with a

marker of syncytiotrophoblast, β HCG, in human placenta (Revised Figure 1A, B). The results of these experiments suggest that both p16 and p21 are expressed in syncytiotrophoblast cells. Owing that p16 is one of the main markers of senescence, these results strongly support our hypothesis that syncytiotrophoblasts are senescent. While expression of p16 was exclusive to syncytiotrophoblasts, we observed p21 expression in other cells types to some extent. Since p21 might play multiple roles, in different cellular processes during development, expression of p21 by itself in different cells does not suggest that these cells are senescent. These findings are now described in the Results section, under "senescence pathways are downregulated in the human placenta during IUGR pathology".

Placentas from 4 uncomplicated and 4 IUGR pregnancies were assessed; but there are more data points in the quantification graph. Please clarify what each data point represents.

Each data point represents a field of view, derived from a section of human normal or IUGR placenta. At least 12 fields of view were quantified from each of normal or IUGR placentas. A detailed explanation of the quantification was added to "Materials and Methods" section under "Human Tissue Collection and Analysis".

3. The authors used knockout mice for p16, 19 and 53, to investigate the hypothesis that the senescence associated pathways are required for proper placental development and function. However, these are complete knockouts, that lack these regulators in all tissues. It is therefore not clear how much indirect effects from other cell types contribute to the phenotype. Please address.

We cannot exclude any indirect effects, which may result from the use of complete knockouts. We clarified this point in the discussion. It is a complex task to dissect such direct and indirect effects in vivo, which will be addressed in future studies.

Which other cell types in the mouse placentas express them?

We performed immunofluorescence co-stainings of the murine syncytiotrophoblast marker (GCM1), together with senescence markers p53 and p19 (ARF) (supplementary Figure S6). We find co-localization of the positive expression of p53 and ARF in the labyrinth syncytiotrophoblast cells (supplementary Figure S6 A,B). This observation further supports that these cells express a combination of senescence markers. It is now referred to in the "Results" section of the manuscript, under "Molecular mechanisms of senescence sustain placental morphology".

4. The authors show that genetic ablation of factors associated with senescence is correlated with changes in the SI detected by DCE-MRI, which evaluates in vivo maternal circulation in the placenta. What about defects in fetal vasculature in the placenta?

In the original study, we performed image analysis of H&E stained sections of mouse placentas from embryos of different genotypes. The evaluation of the pathology was performed by certified mouse pathologists (S.A.Y. and A.dB. on the authors list). The analysis identified consistently collapsed vasculature in the placenta labyrinth of *Cdkn2a* and *cdkn2a;p53* knockout embryos (Figure 3B,C). To further explore this observation, following the reviewers comment, we now performed Immunofluorescence staining of the endothelial cells in the murine placenta of wild-type and *Cdkn2a;p53* knockout mice, using the anti-CD31 antibody (new supplementary figure S3A,B), and the cytotrophoblast marker *Epcam* (see new supplementary figure S3C,D). We observed that *Cdkn2a;p53* knockout placentas have smaller blood vessel lumina (CD31) in the labyrinth. The blood vessels appear compressed compared to wild-type placenta, most likely due to the hyperplasia of the surrounding labyrinth trophoblast cells. In addition, the blood vessels of the *Cdkn2a;p53* knockout placentas are not equally distributed compared to the wild-type vessels in the labyrinth.

The characterization of the differences and similarities in trophoblast histology and placental vascularization between wild-type and mutants by H&E alone (Fig 3 A-E) is limited, and difficult to interpret. For instance, the authors report no significant changes in the *Cdkn1a* null placenta (Fig 3E); however, the placental architecture appears different from wild-type. More definition can be displayed by immunostaining for specific cell types. Please stain for trophoblast and endothelial specific markers to show better the placental structure and changes in labyrinth size, morphology etc.

H&E stainings of the murine wild-type and *CDKn2a;p53* knockout placenta show changes in placenta structure. The trophospongium layer in the *Cdkn2a;p53* knockout placenta is significantly thicker and appears more cellular (see new supplementary figure S4). In addition, we performed immunofluorescence co-staining of cytotrophoblast cells, using the cytotrophoblast marker *Epcam*, and endothelial cells (CD31) of the murine wild-type and *Cdkn2a;p53* knockout placenta. Our observations regarding changes in fetal vasculature are described above. Furthermore, we find that there is more intense *Epcam* staining, most likely illustrating the hyperplasia of the labyrinth trophoblasts (new supplementary figure S3C, D).

Also, it has been shown that syncytiotrophoblasts in the mouse placenta develop from proliferative *Epcam*⁺ labyrinth progenitor cells (LaTP, Ueno et al. Dev Cell 2013) that disappear between E12.5-14.5. If these knockout placentas show hyperproliferative phenotype, is this linked to persistent LaTP proliferation?

This is an interesting point. To answer this question we performed immunofluorescence co-staining of *Epcam* and Ki67 in the placenta labyrinth of wild-type and *Cdkn2a;p53* knockout placentas (see new supplementary figure S7). We find that *Epcam* positive cells express significantly more ki67 in the *Cdkn2a;p53* knockout placenta compared to wild-type (see supplementary S7B). Therefore, labyrinth progenitor cells are more proliferative in the *Cdkn2a;p53* knockout placentas. In addition WB analysis of wild-type and

Cdkn2a;p53 knockout placenta extracts revealed the presence of high amount of Epcam in the knockout placentas (supplementary figure S7C) . This further supports increased proliferation and amount of labyrinth progenitor cells in the Cdkn2a;p53 knockout placentas.

The authors state increased hypercellularity in the labyrinth zones of Cdkn2a null and double Cdkna2a;p53 null placentas. How was cellularity assessed?

Analysis of H&E stained sections of mouse placentas from embryos of different genotypes was performed by certified mouse pathologists (S.A.Y. and A.dB.). Hypercellularity was assessed by image analysis of H&E stained sections.

5. Also, The authors show an increase in cellular proliferation marker Ki67. It is not clear what cells is KI67 signal coming from? It would be helpful to show KI67 with additional placental markers for SynT, LaTP, vasculature etc. BRDU incorporation in the knockout placentas would also be helpful. Assessment of different developmental stages would also be helpful.

As described above, we performed immunofluorescence co-staining of Epcam and Ki67 in the placenta labyrinth of wild-type and Cdkn2a;p53 knockout placentas (see new supplementary figure S7). Quantification of the Epcam+Ki67+ cell fraction showed a significant elevation of this fraction in the Cdkn2a;p53 knockout placenta, compared to wild-type ($17.99\% \pm 1.393$ versus $25.02\% \pm 1.227$, for wild-type and Cdkn2a; p53-knockout, respectively) (Supplementary Figure S7B). In addition, Immunoblot analysis demonstrated an elevation in Epcam protein in the Cdkn2a;p53 knockouts compared to wild-type placentas (Supplementary Figure S7C).

6. This manuscript lacks discussion of differences/similarities of human and mouse placentas, which would be important if results from knockout studies are extrapolated to humans. As an example, in mice, proliferative trophoblasts disappear by E 14.5, but in human, cytotrophoblasts continue to proliferate and differentiate to syncytiotrophoblasts.

A detailed description of the differences between mouse and human placenta was added to the discussion (see Page 17).

Also it would be important to discuss the developmental timing when the senescence pathways are proposed to influence syncytiotrophoblast development in mice and human.

We propose that senescence is induced by cell fusion of cytotrophoblast into syncytiotrophoblast. Since this process occurs at the early stages of placenta formation and throughout pregnancy, we expect to find senescence induction as early as the first trimester. We believe that while induction of senescence pathways might happen early during embryonic development the consequences of their disruption might not be seen during early development and can be observed only later, when the exchange of nutrients

and oxygen through the placenta is increased with the increase in the size of the embryo. Indeed, the lack of cell fusion in syncytin knockout mice, leads to changes in placenta structure and function only starting from E14.5, while fusion itself starts at an earlier stage. Along the same lines, in human placenta pathology of IUGR, while defects in the pathways might occur earlier during embryonic development, their manifestation is only detected when the embryo reaches a certain gestational age. We elaborated on this point in the "Discussion" section.

7. IUGR is associated with placentas that are smaller and with altered chorionic villi architecture and cell composition. Do the authors propose that IUGR placentas are hyperproliferative? How does this result in smaller placentas?

The mice placenta is different than the human placenta. There is no labyrinth in the human placenta but many chorionic villi. The mice placenta is a hemochorial placenta. In placental mediated IUGR, the placenta morphology is influenced by the hypoxic condition, or any other insult that caused placental insufficiency. While in mice placenta it can be expressed by a hyperproliferative labyrinth, similar insults in human placental are leading to hypomature and smaller villi. The syncytial knots, which express many nuclei condensed together and are more prominent in cases of IUGR, might reflect a similar reaction as the hyperproliferative phenotype of the labyrinth in the mice placenta under similar conditions.

8. Another group reported that p53 expression was predominantly present in cytotrophoblasts and increased instead of decreased in IUGR placentas (Levy et al, Amer J Ob Gyn, 2002). Can the authors comment on this discrepancy?

In the study by Levy et al they report enhanced apoptosis correlated with upregulation of p53 primarily in the cytotrophoblast population, while we focus our analysis primarily on syncytiotrophoblast. We have not tested for apoptosis in our study and have not observed an increase in p53 in cytotrophoblasts in our specimens. I believe that there might be a high degree of variability between human patients due to different genetic background and other individual differences or causes of the growth restriction. One possibility is that the cytotrophoblast die due to inability to fuse into syncytiotrophoblast under some circumstances, thus leading to a reduction in the relative amount of syncytiotrophoblast and in the senescence markers expressed in it, which would be consistent with the results of our study. Due to differences in the approaches taken in the two studies it is very hard to compare the results directly or to relate to such a comparison in the manuscript.

9. If possible, provide clinical characteristics of human pregnancies, namely, gestational age at delivery and birth weight, and state how IUGR was defined (including nomogram used to specify weight percentile).

A table of clinical characteristics and all detailed information that was provided to us is now added to the supplementary material (see Table S1)

10. Provide the references for the specific animal strains used.

All the references for the strains that were used are now added to the "Materials and Methods" section of the manuscripts (under "Animals").

Given that some knockout strains are fertile, please clarify whether heterozygous or null mice were interbred to generate null and control (wild-type or heterozygous) littermates. If interbreeding of heterozygous parents was used, how were the embryonic genotypes assessed in order to match with DCE-MRI phenotypes?

Interbreeding of heterozygous parents was used. The embryo positioning inside the female during DCE-MRI scan was noted and matched to the embryos, extracted after euthazation of the pregnant female. The embryonic genotypes were assessed by sampling from each embryo using standard PCR based genotyping protocol, used for the genotyping of the parents. This genotyping procedure is now described in the "Materials and Methods" section (under "In-vivo Contrast-Enhanced MRI Studies" and "Animals").

Referee #3 (Remarks to the Author):

The authors previously showed that cell fusion induces senescence in syncytiotrophoblasts in placenta. In the present study, they show evidence that this senescence phenotype is important for placental structure and function, mainly using mouse genetics and human samples. This is highly original and interesting study. Although the implication of senescence in placental function is correlative, the data seem solid and the study open a new research question.

Fig. 1: they first show a low frequency of senescence markers in IUGR placenta, raising an interesting possibility of functional link between syncytiotrophoblast senescence and placental function. In the IHC figs, the difference is unclear to me. e.g. p53 seems to be higher in IUGR? It would be helpful if they provide more detailed description of histology particularly for non-experts.

We thank the reviewer for pointing this out. A detailed description of histology that describes the different cell types in the placenta is now provided in the supplementary section (new supplementary figure S10). The representative image of p53 IHC in IUGR was now replaced with a more clear image (see revised Figure 1F). Explanation of the method of image analysis is now clarified in "Materials and Methods" section (under "Human tissue collection and analysis").

Fig. 2B: In DCE-MRI, what does the 'recovery phase' correspond to? Again it would be helpful with some explanation for general audience.

Previous studies have shown that the initial loss of contrast enhancement is due to a phagocytic uptake of the contrast material by the syncytiotrophoblasts cells of the labyrinth and trophoblast giant cells. This uptake leads to a subsequent signal intensity reduction. The 'recovery phase' is the re-accumulation of the contrast material in the maternal blood circulation, which leads to a signal intensity (SI) enhancement and is due to recycling of the biotinylated contrast media back into the maternal circulation. This phenomena was first described in a paper by Plaks, V. et.al, Mol Imaging Biol. 2011. We now added an explanation to the Result section of the manuscript (under "molecular mechanisms of senescence affect placental DCE-MRI dynamics").

Transcriptomic analysis: they define differentially expressed genes based on '2-fold change'. I wonder if they consider any statistics, e.g. fdr?

We performed the statistical approach of one-way analysis of variance (ANOVA) to identify differentially expressed genes of 1.8-fold criteria, with statistical significance of $P < 0.05$. The list of up- and down-regulated genes, with their corresponding fold change and p-values is now added to the supplementary (Appendix Tables S2, S3). The description of the statistical criteria used was added to the "Supplementary methods" section (under "microarray analysis").

Fig. 4d, they show GSEA plots. I understand they have used msigdb, but they should specify exactly which gene signatures they have used for each plot, and provide citation associated with them if applicable.

Fig4D: We have now added a table of the GSEA groups that were used in our analysis, with their related links to GSEA (Supplementary Table S4). Additionally, the full list of genes for each group can be obtained from the supplementary (Appendix file F1).

Do they see any phenotype in fetus / fertility in those mutant mice? This would be important to actually show a link between senescence (or p16/p63) and placental functions.

This is an interesting point. We did in fact encounter infertility problems in the *Cdkn2a/p53* knockout mice. Interestingly, previous studies have documented lower birth rate in these mice compared to their wild-type littermates, however, not linked to placenta dysfunction (Krizhanovsky V. et al, Cell, 2008). Previous fertility studies have shown that p53 loss in female, but not in male mice significantly decreases fertility. p53 knockout female mice have much lower pregnancy rates and smaller litter size due to regulation of maternal reproduction mediated by LIF, which is a p53 target gene (Feng Z. et al, FASEB J. ,2011). Our study actually demonstrates a correlation between these genes (p53, *Cdkn2a*) with placental function, owing that DCE-MRI signal intensity dynamics profile of the placenta was impaired in embryo knockouts of these genes.

The implication of senescence deficiency in human IUGR is interesting (Fig. 1 and Fig. 5). Have they also checked proliferation in IUGR syncytiotrophoblasts in placenta?

We find a significant increase in proliferation of the cytotrophoblast population in the senescence *cdkn2a;p53* knockout mice (see new supplementary figure S7). One possibility, is that in human placenta, syncytial knots which express many nuclei condensed together and are more prominent in cases of IUGR, might reflect a similar reaction as the hyperproliferative phenotype of the labyrinth in the mice placenta under similar conditions. Furthermore, previous studies have shown that IUGR placentas exhibit increased signs of impaired telomere homeostasis (Biron-Shental T., et al, Placenta, 2016). We propose that in IUGR the cytotrophoblast cell population undergo increased proliferation as a compensatory mechanism to maintain the integrity of compromised syncytiotrophoblast. We refer to this issue in the discussion section of the manuscript.

Also, I wonder if there is any evidence in the literature for developmental defects in KO mice for e.g. gelatinases, which might be due to placental issues? Although not definitive, these pieces of information would support the correlation between senescence/sasp and the placental function.

Interestingly, several studies have shown a link between lack of gelatinase activity and infertility. A previous study has shown that MMP9 deficiency in mice leads to early manifestation of severe pregnancy pathologies of preeclampsia and IUGR due to impaired trophoblast differentiation and placentation (Plaks V. et al, PNAS, 2013). In addition, fetal single-nucleotide polymorphisms in the MMP2 and MMP9 genes are associated with increased risk of IUGR (Gremlich S. et. al., J Reprod Immunol, 2007). We refer to the notion of gelatinase deficiency with respect to placenta function in the discussion section of the manuscript.

Minor points:

It would be advisable to provide more detailed information about antibodies used, catalogue numbers, dilutions.

A list of antibodies that were used with their related information is now added to the supplementary (Table S5).

Immunoblots in placenta samples: based on Methods, they use whole placenta lysates. Would this be sensitive enough to detect syncytiotrophoblasts-specific changes or do they see p16/p21/p53 expression in other parts of placenta as well (e.g. Fig. 1E)? Or do they somehow enrich syncytiotrophoblasts during sampling?

Immunofluorescence analysis demonstrates that other cells types may also express senescence regulators, such as p21 in human placenta, (Revised Figure 1B).

Nevertheless, within a normal murine placenta, the labyrinth is the largest layer and forms the most substantial fraction, taking the majority of the volume of the placenta to support fetal growth (Woods L. et al, Front Endocrinol, 2018). Therefore, we assumed that whole placenta lysates would be able to detect significant changes in the syncytiotrophoblast, without the need to employ enrichment techniques of the syncytiotrophoblast population.

2nd Editorial Decision

2nd Jul 2019

Thank you for submitting a revised version of your manuscript. It has now been seen by both of the original referees whose comments are shown below.

I apologize for this unusual delay in getting back to you. It took longer than anticipated to receive the referee comments.

As you will see, the referee finds that all criticisms have been sufficiently addressed and recommend the manuscript for publication. However, before I can accept the manuscript, there are a few editorial issues I need you to address.

REFeree REPORTS:

Referee #2:

The authors have performed extensive revisions which include additional experiments and improved presentation and discussion of the data. The manuscript has improved greatly, and I would be happy to recommend publication of this work.

Referee #3:

the authors have addressed reviewers' questions adequately.

2nd Revision - authors' response

12th Jul 2019

The authors performed the requested editorial changes.

3rd Editorial Decision

26th Jul 2019

Thank you for submitting your revised manuscript. I have now looked at everything and all looks fine. Therefore I am very pleased to inform you that your manuscript has been accepted for publication in The EMBO Journal.